# AnyExpress: One Adapter Enabling Highly Flexible Audio-Driven Portrait Animation

## Abstract

Portrait animation, particularly audio-driven portrait animation, requires flexibility in facial expressions, head movement, and dynamic contexts. However, existing diffusion-based methods rely heavily on the design of ReferenceNet, leading to increased training complexity and incompatibility with other custom base models or adapters, also limiting face position, view changes, and animated context generation. To address these challenges, we propose *AnyExpress*, a lightweight, modular framework that eliminates the need for ReferenceNet, reducing the number of trainable parameters by *7* times. By training one plug-and-play *audio-motion adapter*, it allows freeform, expressive audio-driven portrait animation with any face pose and any animated context, while supporting text-driven modifications. In the context of character generation, there are two primary methods to control the desired character attributes. First, if a specific ID needs to be assigned, this can be achieved through ID controls (e.g., IP-Adapter-Face). Alternatively, the character's attributes can be controlled through textual descriptions. Through comprehensive qualitative and quantitative analyses, *AnyExpress* demonstrates unprecedented freedom in generating videos with dynamic background, lower training demand, and seamless integration with evolving custom models and control adapters, providing a flexible solution for diverse generation needs. The demo is available at https://anyexpress-alpha.github.io/Any, and we will release our code, encouraging further improvement.

## 1 Introduction

The human face talking video is not merely a static object in the context of multimedia and communication; rather, it is a dynamic, vibrant object in which the background changes and the speaker's face can appear from front-facing to side profiles and other poses (Wang et al., 2021a;b; Zhong et al., 2023b; Zhang et al., 2023a). Recently, advanced methods built on ReferenceNet-based diffusion frameworks (Hu, 2024) have demonstrated superior performance in audio-driven portrait animation (Chen et al., 2024; Tian et al., 2024; Wang et al., 2024a; Wei et al., 2024; Xu et al., 2024a).

However, we argue that this design has two limitations. First, as shown in Fig. 1 middle top, reliance on a ReferenceNet increases the model's parameter count and complicates the training procedure. This added 2D-UNet results in a highly coupled framework, additional training stages, and greater resource requirements, making the approach less scalable and generalizable under other control adapters. Second, as shown in Fig. 1 right top, current methods overly constrain video outputs, with the pose tightly bound to the reference image, limiting view change through large angles or outpainting generation. Additionally, these methods can only produce a static background as the reference image, which fails to reflect the dynamic environments of the real-world.

Considering this, we argue that highly flexible, audio-driven talking face generation is crucial for real-world applications. We term this *Freeform Portrait Animation*, which aims to generate talking faces in any configuration (Fig. 1 right): ❶ **Any face pose**, allowing for faces to be generated from any angle or position, extending beyond the reference image; ❷ **Any animated context**, seamlessly integrating talking faces with animated elements and backgrounds; ❸ **Any control via text instructions**, enabling users to modify background and identity attributes through text prompts.

To address these limitations and achieve *Freeform Portrait Animation*, we design a solution centered on the principle of "*Low Coupling, High Cohesion*", enabling seamless integration with other

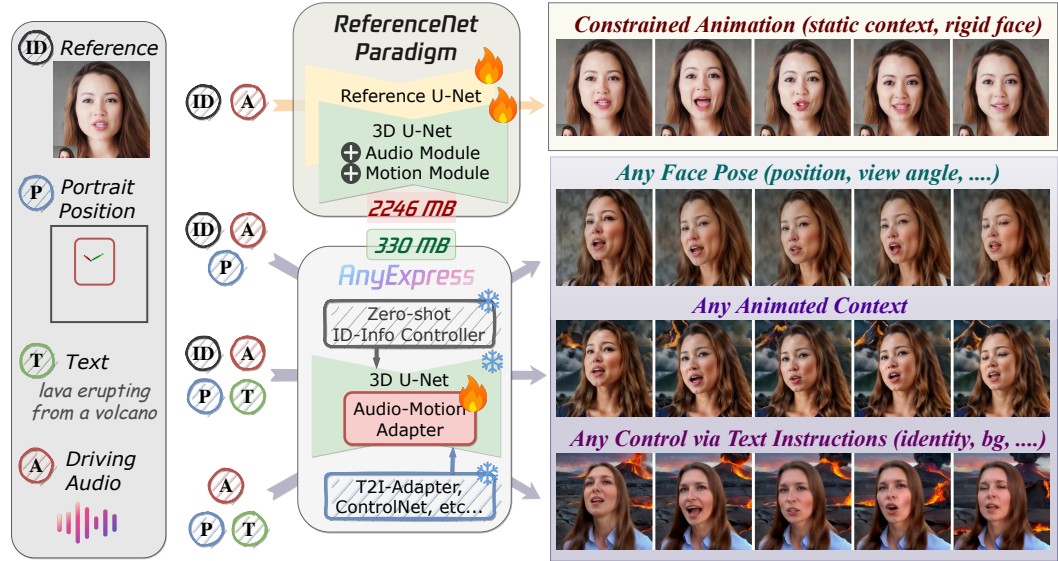

Figure 1: Left: control signals (identity, portrait position, text, audio) used in audio-driven portrait animation. Middle: ReferenceNet paradigm (highly coupled, large model with constrained animation) and our ***AnyExpress*** (lightweight, plug-and-play framework with flexible controls). Right: the *Freeform Portrait Animation* task, enabling any face poses, animated contexts, and text control.

personalized diffusion models and compatibility with additional adapters for various control conditions. To achieve this, we propose to train a self-contained "*Audio-Motion Adapter*" with a single, well-defined purpose: learning audio-driven portrait animation priors and aligning them with the internal knowledge of text-to-image(T2I) models. This design overcomes the limited generalizability of the ReferenceNet-based paradigm and eliminates the need to duplicate knowledge across modules, particularly ReferenceNet. The audio-motion adapter is optimized not only for lip-sync and speech animation but also for preserving the intrinsic capabilities of video diffusion models (Ho et al., 2022). These capabilities include dynamic background animations, unrestricted movement and positioning of the subject, and generating more flexible and realistic talking face animations.

Given these limitations and design objectives, we propose ***AnyExpress***, a scalable audio-driven portrait animation framework featuring two key innovations. **1) ReferenceNet-Free**: Unlike prior methods, AnyExpress eliminates the reliance on ReferenceNet and its excessively strong control. Instead, it uses an optional, weak but flexible Face-ID control signal to maintain consistency of a target identity across frames, reducing computational load while still preserving facial features. **2) Audio-Motion Adapter**: This modular adapter focuses solely on motion dynamics and controls the generation of talking faces based on audio input without retraining the entire UNet architecture. Its modularity allows seamless integration with various custom T2I models and other adapters. Furthermore, we extend a *Progressive Prefix Conditioning* strategy under the ReferenceNet-Free paradigm, to ensure smooth transitions between segments in long video sequences by inheriting prefix frames, thus maintaining continuity in motion and appearance. Our contributions are summarized as:

- *Limitations Identification*: We identify the limitations of current audio-driven portrait animation methods, particularly their reliance on ReferenceNet, which adds computational complexity and limits flexibility in face poses and backgrounds. To address this, we introduce *Freeform Portrait Animation*, a task focused on greater flexibility and real-world applicability, and to the best of our knowledge, we are the first to introduce this task.

- *Methodology*: We present ***AnyExpress***, a plug-and-play solution for flexible portrait animation using a lightweight *Audio-Motion Adapter*. This adapter efficiently handles motion dynamics and lip-sync, seamlessly integrating with personalized diffusion models and supporting various control conditions (e.g., IP-Adapter-Face (Ye et al., 2023), text instructions).

- *Superiority*: Extensive experiments show that our method generates flexible, high-quality portrait animation videos across diverse conditions, while maintaining satisfactory identity consistency. These conditions can be easily combined for multi-condition control without further fine-tuning.

## 2 RELATED WORK

**Diffusion-Based Portrait Animation.** Recent diffusion-based portrait animation methods focus on different control modalities—audio, visual signals, or multi-modal combinations. *Audio-driven* methods synchronize lip movements and head motions but struggle with large-scale head movements, limiting expressiveness (Wang et al., 2021a; Yu et al., 2023; Yang et al., 2023; Zhang et al., 2023c). *Visual-driven* methods use facial landmarks and poses to guide animations but fail to incorporate audio dynamics and often introduce distortions when driving signals differ significantly from the reference image in identity or proportions (Xie et al., 2024; Ma et al., 2024). Several *multi-modal* methods attempt to combine audio and visual signals for more nuanced control (Drobyshev et al., 2024; Wang et al., 2024a; Yang et al., 2024a). However, these existing methods primarily rely on strong control via ReferenceNet, resulting in video generation being overly constrained by the reference image. This not only limits flexibility in adapting to dynamic external conditions, such as backgrounds or facial poses, but also complicates the training processes and prevents adaptive evolution alongside new developments in diffusion models (Chen et al., 2023a; Sauer et al., 2023; 2024; Esser et al., 2024). In this work, we address these limitations by introducing a lightweight adapter that supports *Freeform Portrait Animation*. Our approach enables face generation from various angles and positions within dynamic environments, providing a more flexible and natural animation experience.

**Adapter.** Adapters were introduced to make fine-tuning large pre-trained models more efficient, enabling transfer learning with compact modules instead of full-model fine-tuning (Houlsby et al., 2019; Li et al., 2022; Chen et al., 2023b; Hu et al., 2021). As large-scale datasets have grown (Schuhmann et al., 2022), diffusion models now have billions of parameters. Fine-tuning all these for each task (Rombach et al., 2022; Peebles & Xie, 2023; Podell et al., 2024; BlackForestLabs, 2024; Yang et al., 2024c) is computationally expensive and can cause issues like catastrophic forgetting (Smith et al., 2024; Gao & Liu, 2023), where models lose previously learned knowledge. To address this, adapter-based methods (Mou et al., 2024; Zhang et al., 2023b; Zhong et al., 2023a; Xing et al., 2024) insert lightweight modules into diffusion models for task-specific adaptation. However, current portrait animation methods heavily rely on fine-tuning the entire diffusion model, with some approaches even replicating the UNet, which dramatically increases computation and hinders transferability, highlighting the necessity of lightweight and scalable solutions.

## 3 METHOD

### 3.1 PREREQUISITE

**Latent Diffusion Models.** Latent diffusion models represent a class of diffusion models that operate within the encoded latent space produced by an autoencoder (Van Den Oord et al., 2017), which converts images $\mathbf{X}_0$ into latent representation $\mathbf{z}_0 \in \mathbb{R}^{H_z \times W_z \times D_z}$. In this work, we choose Stable Diffusion (SD) (Rombach et al., 2022) as our base model, which incorporates condition embeddings $\mathbf{c} \in \mathbb{R}^{D_c}$ during the diffusion process. The training objective for Stable Diffusion is encapsulated by:

$$\mathcal{L} = \mathbb{E}_{\mathbf{z}_t, \mathbf{c}, \epsilon \sim \mathcal{N}(0,1), t} \left[ \|\epsilon - \epsilon_\theta(\mathbf{z}_t, t, \mathbf{c})\|_2^2 \right],$$

with $t$ uniformly sampled from $\{1, \ldots, T\}$. Here, $\epsilon_\theta$ denotes the denoising U-Net, which includes Spatial Transformer layers that facilitate both self-attention and cross-attention.

**Cross Attention as Condition Guidance.** The cross-attention in U-Net ensures that generated images are contextually aligned with the input auxiliary conditions, which can be expressed as:

$$\text{CrossAttn}(\mathbf{z}_t, \mathbf{c}) = \text{Attention}(\mathbf{Q}_z, \mathbf{K}_c, \mathbf{V}_c), \tag{1}$$

$$\text{with} \quad \mathbf{Q}_z = \mathbf{W}_Q \mathbf{z}_t, \mathbf{K}_c = \mathbf{W}_K \mathbf{c}, \mathbf{V}_c = \mathbf{W}_V \mathbf{c}, \tag{2}$$

where $\mathbf{W}_Q$, $\mathbf{W}_K$, and $\mathbf{W}_V$ are learnable projection matrices. For the choice of conditions $\mathbf{c}$, SD employs a CLIP text encoder (Radford et al., 2021) to transform the input text prompt into a conditional text embedding $\mathbf{c}_{\text{text}}$. For portrait animation, a pretrained Wav2Vec encoder (Baevski et al., 2020) is usually utilized to encode audio into condition embeddings $\mathbf{c}_{\text{aud}}$.

**Model Architecture.** Previous methods for audio-driven portrait animation typically rely on the ReferenceNet-based framework (*Appendix B*) with tightly coupled, jointly trained modules, which

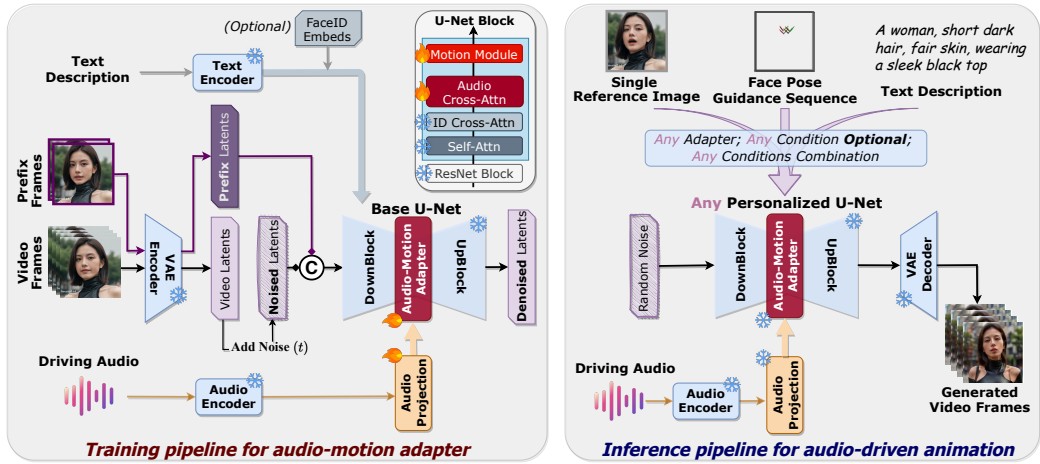

Figure 2: Left: Only a lightweight, self-contained *Audio-Motion Adapter* is fine-tuned with prefix frames and driving audio for smooth transitions. Right: The adapter allows seamless integration with personalized T2I models and compatibility with various control conditions (*e.g.*, face keypoints, text instructions). This removes redundant modules from previous methods, supporting audio-driven portrait animation while preserving the creativity and generalizability of diffusion models.

complicates training processes and limits flexibility. These modules include: *Control Modules* ($\mathcal{C}$) for additional control signals like keypoints, face mesh, and audio; a *Motion Module* ($\mathcal{M}$) to ensure cross-frame coherence; a *Reference 2D U-Net* ($\mathcal{R}$) that extracts reference image features as a strong control for the backbone 3D U-Net; a *Backbone 3D U-Net* ($\mathcal{B}$) to integrate all control signals. However, this coupled framework struggles with personalized tasks like style transfer, as the U-Net cannot be easily replaced with a personalized T2I model, and adapting it requires costly retraining.

We believe a lightweight adapter model can effectively address these challenges by aligning control signals with the existing knowledge of T2I models, without the need to learn new generative abilities. To solve this "alignment" issue, the proposed **AnyExpress** eliminates the need for joint training across all modules. Instead, it focuses on an *Audio Control Module* ($\mathcal{C}_{\text{aud}}$) and a *Motion Module* ($\mathcal{M}$), forming the "Audio-Motion Adapter" (Fig. 2, left). This design enables the adapter to learn the necessary audio-driven portrait animation priors, allowing it to animate any personalized T2I model without additional training (Fig. 2, right), making it a versatile, one-time solution for various tasks.

### 3.2 LESS IS MORE: REFERENCENET LIMITS GENERATION FLEXIBILITY

In portrait animation, prior methods rely heavily on ReferenceNet to ensure identity consistency by imitating the reference image. However, this severely restricts generation freedom, limiting facial expressions, head movements, and background dynamics. Moreover, without further joint training, these methods struggle to adapt to out-of-domain control signals such as text prompts or face keypoints. In this work, we challenge this reliance on ReferenceNet by replacing it with a weaker control mechanism, Text-FaceID control, which combines text descriptions with face features of reference identities. The Text-FaceID control is formulated as:

$$\text{CrossAttn}_{\text{ID}}(\mathbf{z}_t, \mathbf{c}_{\text{text}}, \mathbf{c}_{\text{id}}) = \text{Attention}(\mathbf{Q}_z, \mathbf{K}_c^{\text{text}}, \mathbf{V}_c^{\text{text}}) + \text{Attention}(\mathbf{Q}_z, \mathbf{K}_c^{\text{id}}, \mathbf{V}_c^{\text{id}}), \quad (3)$$

$$\text{with} \quad \mathbf{K}_c^{\text{text}} = \mathbf{W}_K^{\text{text}} \mathbf{c}_{\text{text}}, \mathbf{V}_c^{\text{text}} = \mathbf{W}_V^{\text{text}} \mathbf{c}_{\text{text}}, \mathbf{K}_c^{\text{id}} = \mathbf{W}_K^{\text{id}} \mathbf{c}_{\text{id}}, \mathbf{V}_c^{\text{id}} = \mathbf{W}_V^{\text{id}} \mathbf{c}_{\text{id}}, \quad (4)$$

where, $\mathbf{c}_{\text{id}}$ is the face features of the reference identity; $\mathbf{W}_K^{\text{text}}$, $\mathbf{W}_V^{\text{text}}$, $\mathbf{W}_K^{\text{id}}$, and $\mathbf{W}_V^{\text{id}}$ are pretrained projection matrices. In this work, we choose IP-Adapter-Face (Ye et al., 2023) as the identity controller to initialize $\mathbf{W}_K^{\text{id}}$ and $\mathbf{W}_V^{\text{id}}$, and keep them frozen throughout.

In Fig. 3, we compare the exploration behavior of *Strong Control* (ReferenceNet) with *Weak Control* (Text-FaceID) in U-Net self-attention heads. This comparison aims to highlight the differences in diversity and focus among attention heads, which are critical for supporting *Freeform Portrait Animation*. We randomly sampled 30 identities and 50 audio clips from evaluation datasets, performing inference with all other settings identical to compare strong and weak control. By examining

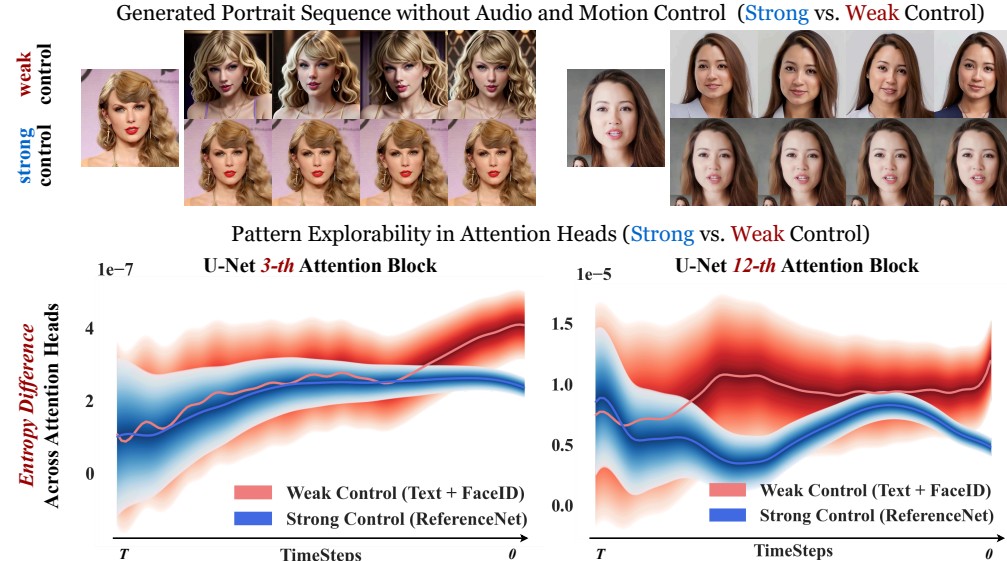

Figure 3: **Strong vs. Weak Control in Portrait Animation**. Top: portrait sequences generated with identity control only, with Weak Control enabling more expressive and varied outputs. Bottom: entropy difference across attention heads for the 3rd (shallow) and 12th (deep) U-Net blocks, representing high- and low-level semantics. The solid line represents the average of the distribution.

the entropy difference across attention heads, we plot the distribution of all samples as shown in Fig. 3, providing insights into how these control methods encourage exploration and adaptability during the generation process: 1) **Early Plateau of Strong Control (ReferenceNet)**: Strong control (blue) plateaus early, indicating that the attention heads quickly converge on similar patterns, rigidly focusing on the reference image. This leads to reduced attention diversity, limiting the model's ability to explore different features and adapt to out-of-domain control signals. 2) **Gradual Increase in Entropy for Weak Control (Text-FaceID)**: In contrast, weak control (red) shows a gradual increase, reflecting more dynamic and adaptive attention heads. This enables the model to explore a broader range of patterns and possibilities throughout the denoising process, resulting in more flexible and varied output. 3) **Larger Exploration Area for Weak Control**: The wider red heatmap in the weak control case demonstrates greater exploration across attention heads. This flexibility supports the generation of more dynamic facial expressions, backgrounds, and better adaptability to additional control signals. Moreover, in the experiments part (Sec. 4), we validate that, without relying on the strong control of ReferenceNet, this shift to weak identity control still maintains identity consistency, offering a more flexible and efficient solution for audio-driven talking face generation.

**Proposition 3.2 (Enhanced Generation Flexibility with Weak Control)** *The Weak Control mechanism enables broader exploration in attention heads compared to Strong Control, resulting in more dynamic, freeform animations and better adaptability to new control signals. We argue that replacing ReferenceNet with Text-FaceID can enhance generative freedom while maintaining identity consistency, paving the way for generating expressive and flexible talking face videos.*

### 3.3 MORE FROM LESS: PROGRESSIVE PREFIX CONDITIONING

Generating long, temporally consistent videos with smooth transitions across extended frames is a significant challenge. *Progressive Fusion* strategy is widely adopted in prior methods (Fig. 4 left), which tackles this by averaging overlapping latents between adjacent windows. However, this naive approach often results in non-smooth transitions at window boundaries, as each window is processed independently, and direct averaging lacks meaningful semantic alignment. In ReferenceNet-based methods, strong control signals minimize variance between frames, making transition issues less noticeable. In contrast, our ReferenceNet-free design, with its greater variance in face poses and

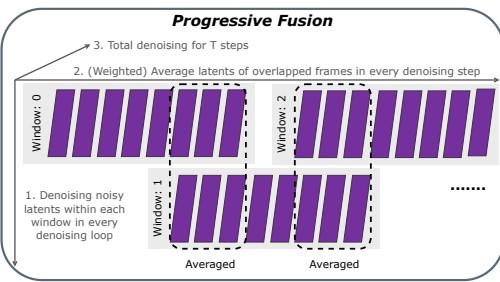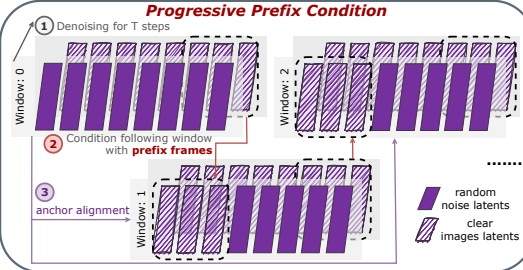

Figure 4: **Comparison of long video generation strategies**. Left: *Progressive Fusion* averages overlapped frames, causing non-smooth transitions. Right: *Progressive Prefix Conditioning* uses prefix frames and anchor alignment for natural, consistent transitions across windows.

backgrounds, amplifies the inconsistencies of the progressive fusion strategy. This leads to unnatural transitions and visible artifacts between windows, as shown in our ablation studies (Sec. 4.4).

To overcome these limitations, we propose *Progressive Prefix Conditioning* (Fig. 4 right). This method builds consistency among windows by conditioning each new window on prefix frames taken from the end of the previous window. This is similar to how, in life, each moment informs the next, creating a cohesive and natural result. The prefix frames serve as a "guiding thread," informing the generation of upcoming frames and ensuring seamless continuity. However, even with prefix frames, deviations in texture, color, and other attributes can still arise across windows. To mitigate this, we further introduce *anchor alignment*. During the denoising of the first window (the anchor), we store the mean and variance of its latents across U-Net blocks at each timestep. In subsequent windows, we align their latents with the stored anchor's statistics at every timestep, ensuring that the generated frames remain consistent with those of the first window. Unlike *Progressive Fusion*, which focuses on static averaging, *Progressive Prefix Conditioning* allows for dynamic refinement and adjustment, promoting smooth transitions. Refer to Algorithm C.1 for the formulated procedure.

### 3.4 HARMONY IN DIVERSITY: A VERSATILE AUDIO-MOTION ADAPTER

**Training.** In our design, we fine-tune only the audio and motion modules, using a two-stage strategy to prevent overfitting in the motion module. Since the temporal module is pretrained and requires less training time than the randomly initialized audio module, fine-tuning both with the same learning schedule can cause the motion module to overfit the downstream training datasets. To address this, in stage I, we perform **longer fine-tuning** of both modules to enable the audio module to drive lip synchronization and ensure smooth transitions. In stage II, we restore the motion module from pretrained weights and apply **shorter fine-tuning** with the audio module from stage I, achieving proper integration of these two modules without identity overfitting.

**Inference.** During inference, a single reference image and driving audio are taken as input to generate an audio-driven portrait animation video. To ensure visual consistency in long videos, we use the last 4 frames of the previous window as the prefix frames for the next (Sec. 3.3). Additionally, the base model parameters remain unchanged throughout. After fine-tuning, the audio-motion adapter can animate any personalized T2I models without needing extra data collection or further training. As shown in Fig. 6b, 11, 12, our design seamlessly supports various functionalities already present in the base model and the broader adapter ecosystem. It can integrate with identity-image control (Ye et al., 2023; Wang et al., 2024b; Guo et al., 2024c; Song et al., 2024), pose control (Zhang et al., 2023b; Mou et al., 2024), and text control. This adaptability allows for versatile multi-modal control without the need for retraining large diffusion models, making it highly scalable for portrait animation tasks.

## 4 EXPERIMENTS

### 4.1 EXPERIMENTAL SETTINGS

**Implementation Details.** The training process used 8 NVIDIA V100 GPUs over two stages: 180,000 steps in stage 1 and 30,000 steps in stage 2, with a batch size of 16 and a video resolution of 512 × 512. Each instance generated 16 video frames, and noise latents are concatenated with the first 4

ground truth frames for prefix conditioning. Both stages had a learning rate of 1e-6, with the motion module initialized using Animatediff weights (Guo et al., 2024b). A 0.1 dropout was applied to reference images, text instructions, and audio during training. For inference, sequence continuity was ensured by concatenating noisy latents with the last 4 motion frames from the previous window.

**Datasets.** *AnyExpress* is trained by HDTF (Zhang et al., 2021) (28k clips, 42.58 hours), and other collected videos (171k clips, 263.23 hours). The facial regions in these videos are cropped and resized to 512×512. The total training dataset comprises approximately 300 hours of video. We cleaned the data by retaining single-person speaking videos with strong lip-audio consistency while excluding those with scene changes, significant camera movements, or excessive noise. Moreover, the quantitative evaluation (Table. 2) was performed on the HDTF, CelebV (Zhu et al., 2022) datasets.

**Evaluation Metrics.** Unlike ReferenceNet-based methods that mainly reconstruct the reference image with minor modifications and rely on metrics like FID and FVD, our *Freeform Portrait Animation* focuses on personalizing or re-contextualizing the talking face identity. Thus, we introduce evaluation metrics tailored to measure the effectiveness for this task. For **portrait animation quality**, *Pose Diversity Score ($\Delta P$)* measures head motion intensity , with higher scores indicating more diverse movements and flexibility (Xu et al., 2024a); and *Sync-C* and *Sync-D* evaluate lip synchronization with the audio, with higher Sync-C and lower Sync-D indicating more natural lip movements (Chung & Zisserman, 2017). For **video quality**, *DOVER Score* (Wu et al., 2022; 2023) represents the overall quality of videos from aesthetic and technical perspectives. For **identity preservation**, *FaceID Consistency* calculates cosine similarity between generated faces and the reference image using a face recognition model, while *CLIP-I Score* (Radford et al., 2021) measures structural similarity to ensure consistent facial features and alignment. Refer to *Appendix* A.2 for details.

Table 1: Comparison of various portrait animation methods and their control freedom.

| Methods | Open-Source | Control Freedom | | | |
|---|---|---|---|---|---|
| | | Audio-Driven | Face Pose | Animated Context | Text-Driven |
| SadTalker (2023c) | ✔ | ✔ | ✘ | ✘ | ✘ |
| Hallo (2024a) | ✔ | ✔ | ✘ | ✘ | ✘ |
| LivePortarit (2024a) | ✔ | ✘ | ✔ | ✘ | ✘ |
| FollowEmo (2024) | ✔ | ✘ | ✔ | ✘ | ✘ |
| X-Portrait (2024) | ✔ | ✘ | ✔ | ✘ | ✘ |
| EMO (2024) | ✘ | ✔ | ✔ | ✘ | ✘ |
| VASA-1 (2024b) | ✘ | ✔ | ✔ | ✘ | ✘ |
| AniPortrait (2024) | ✔ | ✔ | ✔ | ✘ | ✘ |
| MegActor (2024a) | ✔ | ✔ | ✔ | ✘ | ✘ |
| V-Express (2024a) | ✔ | ✔ | ✔ | ✘ | ✘ |
| EchoMimic (2024) | ✔ | ✔ | ✔ | ✘ | ✘ |
| MegActor-$\Sigma$ (2024b) | ✘ | ✔ | ✔ | ✘ | ✘ |
| *AnyExpress* (**Ours**) | ✔ | ✔ | ✔ | ✔ | ✔ |

**Baselines.** Currently, no open-source methods support text instruction control for audio-driven portrait animation. While some methods incorporate face pose control, they do not accommodate the text control needed for our *Freeform Portrait Animation* task. Thus, our comparison focuses on the **Any Face Pose** aspect, where pose control is directly relevant. We selected *AniPortrait*, *MegActor*, *EchoMimic*, and *V-Express* for comparison due to their public availability and support for audio and face control signals, while other methods lack the multi-modal control needed for a fair comparison.

## 4.2 COMPARISON OF ANY FACE POSE GENERATION

In the Any Face Pose task, most previous methods struggle to generate flexible face poses while maintaining identity consistency and video quality (Table 2). When face pose control signals deviate significantly from the reference image, these methods often introduce artifacts, limited outpainting, and non-smooth transitions (Fig. 5). This is due to the excessive strong control imposed by ReferenceNet, which limits the generative flexibility of diffusion models. *V-Express* and *AniPortrait* can extend beyond the face to generate upper body parts like shoulders, but this often introduces artifacts that degrade video quality, reflecting the rigid, unnatural results from strong control. On the other hand, *EchoMimic* and *MegActor*, fail to extend beyond the face, resulting in

Table 2: Performance of audio-driven portrait animation methods under **Any Face Pose** conditions. Bold values indicate the best results, while underlined values denote the second-best results.

| Methods | Params | Portrait Animation | | | Video Quality | ID Preservation | |
|---|---|---|---|---|---|---|---|
| | | ΔP ↑ | Sync-C ↑ | Sync-D ↓ | DOVER Score ↑ | FaceSim ↑ | CLIP-I ↑ |
| V-Express | 2.2B | 0.371 | 6.371 | 8.424 | 0.593 | 0.360 | 0.690 |
| EchoMimic | 2.1B | 0.402 | **6.621** | **8.132** | 0.690 | 0.353 | 0.694 |
| MegActor | 2.1B | 0.411 | 5.745 | 8.923 | 0.657 | 0.386 | 0.745 |
| AniPortrait | 2.5B | **0.438** | 6.303 | 8.541 | 0.762 | 0.418 | 0.786 |
| *AnyExpress* | **0.3B** | 0.425 | 6.552 | 8.397 | **0.804** | **0.453** | **0.812** |

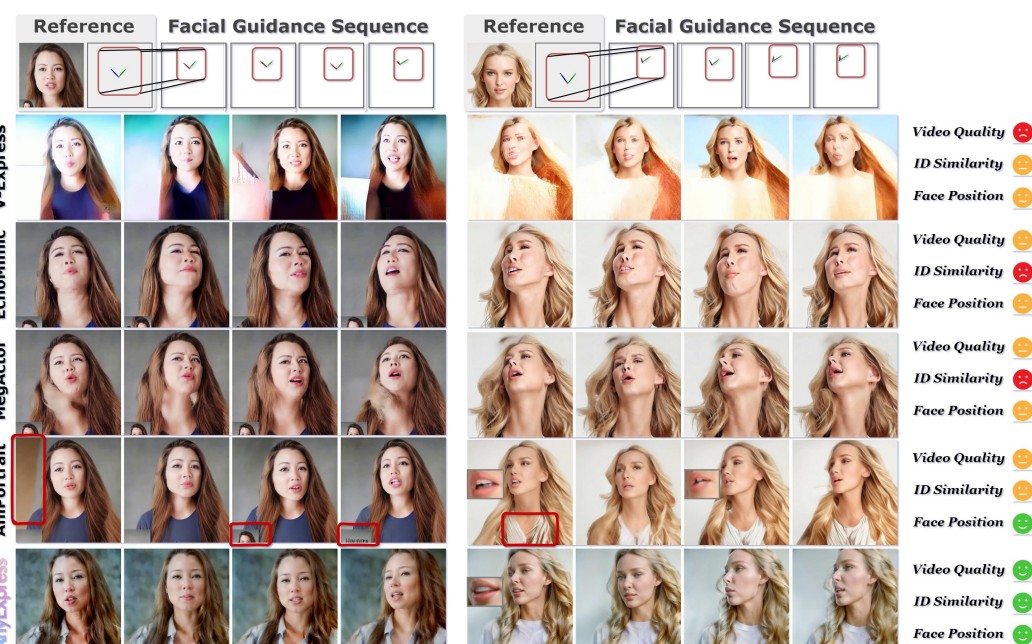

Figure 5: Comparison on **Any Face Pose** generation. ① *V-Express* and *AniPortrait* extend beyond the face but introduce significant artifacts; ② *EchoMimic* and *MegActor* fail to extend upper body movements, resulting in constrained and unsatisfactory facial angles; ③ *AnyExpress* achieves superior face pose flexibility with high video quality and identity preservation.

constrained facial movements and unsatisfactory pose angles, with lower *FaceSim* and *CLIP-I* scores (Table 2). These methods remain tightly bound to the reference image, limiting pose diversity and dynamism. In contrast, *AnyExpress* generates diverse, natural face poses with high video quality and identity preservation. By using a lighter, more flexible control mechanism, *AnyExpress* avoids artifact issues in upper-body outpainting and allows for greater pose and body movement flexibility while maintaining video quality and identity consistency. More results are provided in *Appendix* D.3.

## 4.3 EVALUATION OF ANIMATED CONTEXTS AND TEXT-BASED CONTROL

We evaluate AnyExpress in two areas where traditional methods fall short: animated backgrounds and text-based control, showcasing its unique flexibility and creativity in portrait animation.

**Any Animated Contexts.** Previous methods typically produce static backgrounds that fail to reflect the dynamic environments of the real-world, as they are tightly bound to the reference image. AnyExpress, by contrast, enables the seamless integration of animated backgrounds into portrait animations. As shown in Fig. 6a, high-quality animations are generated where the background elements are dynamic and interact naturally with the foreground subject. See *Appendix* D.4 for more.

**Any Text-Based Control.** A key limitation of existing methods is their inability to handle text prompts for controlling identity or background attributes, while AnyExpress addresses this by maintaining base models' inherent text-based control. As shown in Fig. 6b, AnyExpress: 1) accurately generates

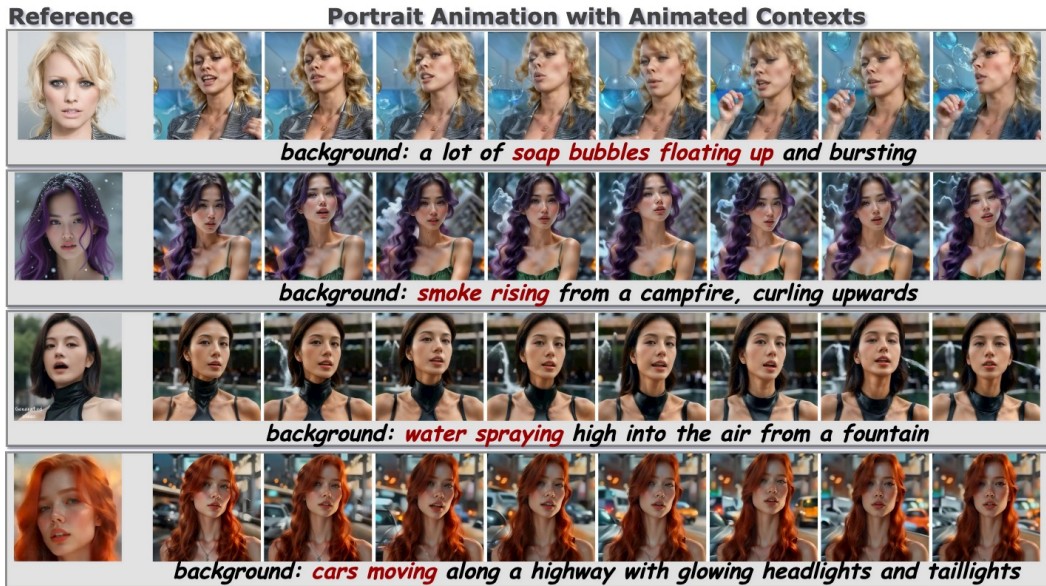

(a) AnyExpress generating portrait animations with **animated contexts**, showcasing its ability to create contextually rich animated environments that enhance the overall animation and realism.

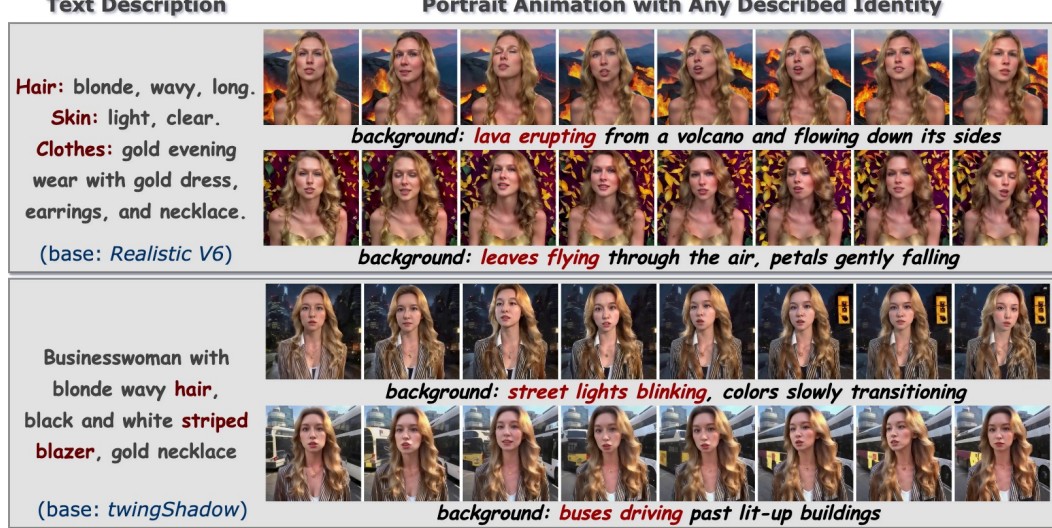

(b) AnyExpress generating portrait animations based solely on **text instructions** (w/o reference image), demonstrating its ability to handle complex text-based control and generate diverse, customized animations.

Figure 6: Evaluation of AnyExpress on **Any Animated Contexts** and **Any Text-Based Control**.

identities from text descriptions; 2) combines identity and background for cohesive animations. The upper and lower rows use a realistic base and an Asian-style model, respectively, demonstrating seamless integration with personalized T2I models. See *Appendix* D.1 and D.5 for more.

## 4.4 ABLATION STUDY

**How Progressive Prefix Conditioning Maintaining Consistency?** As shown in Fig. 7a, the Progressive Fusion method leads to inconsistencies across windows, resulting in non-smooth transitions and misaligned frames. Without anchor alignment, as seen in the middle row, there are visible color deviations and degraded video quality. However, Progressive Prefix Conditioning with anchor alignment (bottom row) significantly improves the temporal consistency and overall visual quality, maintaining color and texture uniformity across windows.

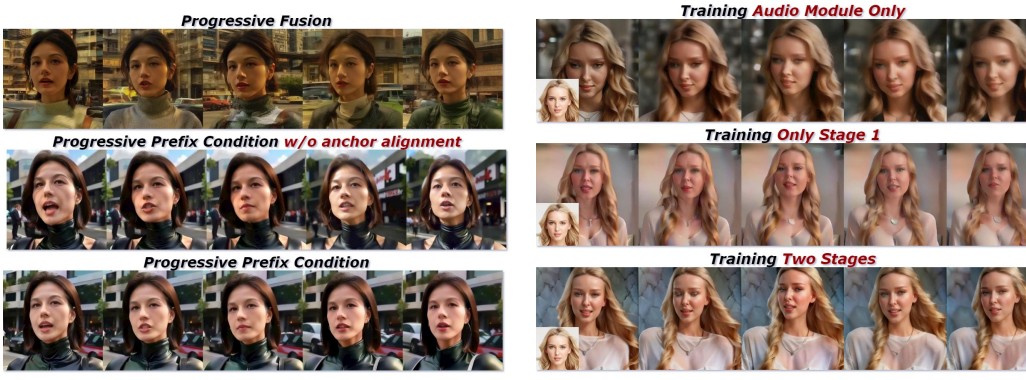

(a) Top: *Progressive Fusion* introduces video inconsistencies across windows; Middle: without *anchor alignment*, there are color deviations and degraded quality.

(b) Top: training the audio module only results in poor face pose control and non-smooth transitions; Middle: only stage 1 leads to identity overfitting.

Figure 7: Ablation studies on *Progressive Prefix Conditioning* and training strategies.

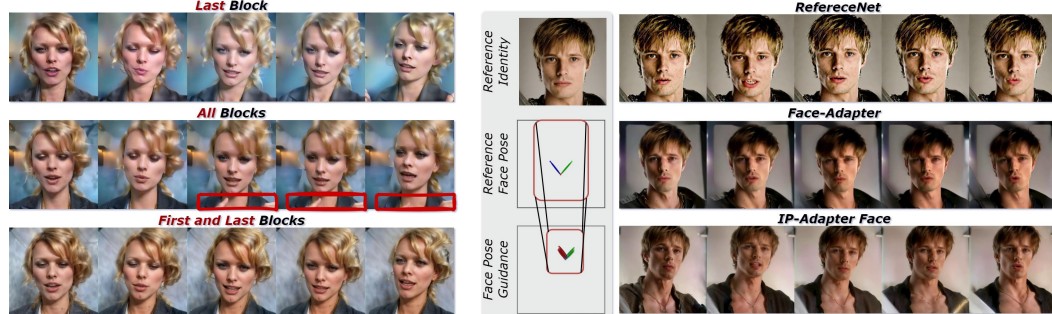

(a) Top: training only the last block of motion module leads to non-smooth transitions; Middle: training all blocks results in identity overfitting.

(b) Strong control from both ReferenceNet and Face-Adapter leads to poor face pose control and limits flexibility, while IP-Adapter-Face allows for more natural pose generation.

Figure 8: Ablation Studies on motion module trainable blocks and the identity information controller.

**Audio and Motion Module Training Strategy.** As shown in Fig. 7b, without fine-tuning the motion module (top), the model fails to align face poses, audio signals and produces non-smooth transitions. Using only stage 1 (middle) leads to identity overfitting and visual inconsistencies. The two-stage strategy (bottom) resolves these issues, ensuring smooth transitions and identity preservation.

**Motion Module Trainable Blocks.** Each U-Net block has one motion module, with three motion blocks in both the encoder and decoder (Fig. 10). As shown in Fig. 8a, training only the last block (top) causes non-smooth transitions due to insufficient trainable parameters. Training all three blocks (middle) leads to identity overfitting, causing visual issues like the "hand problem." Training the first and last blocks (bottom) strikes a balance, achieving smooth transitions without overfitting.

**Identity Information Controller.** In Fig. 8b, replacing the IP-Adapter-Face with a ReferenceNet (top) or Face-Adapter (Han et al., 2024) (middle) shows overly strong control, leading to poor pose control and reduced flexibility, limiting the model's ability to generate natural, dynamic poses.

## 5 CONCLUSION

In this paper, we introduced ***AnyExpress***, a scalable framework for audio-driven portrait animation that eliminates the need for ReferenceNet, reducing computational complexity and enhancing compatibility with custom models. Key innovations include a flexible Face-ID control for identity consistency and an Audio-Motion Adapter for motion dynamics without retraining the entire U-Net architecture. A *Progressive Prefix Conditioning* strategy also ensures smooth transitions in long video sequences. Comprehensive analyses show AnyExpress's ability to generate dynamic, expressive animations with lower training demands.

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

# Part I

# Appendix

## Table of Contents

# A EXTENDED EXPERIMENTAL SETTINGS

## A.1 IMPLEMENTATION DETAILS

All primary experiments are conducted using `Stable Diffusion v1.5`[1], with an image size of 512x512x3 and a latent space of 64x64x4. For personalized models (Fig. 6b), `Realistic V6`[2] and `TWingShadow`[3] are employed. Inference are conducted on a single Nvidia V100 GPU with 30 timesteps. Considering large guidance scale (Ho & Salimans, 2021) often leads to overly saturated and unnatural images. We employ *dynamic thresholding* (Saharia et al., 2022) to handle color issues by clipping outof-range pixel values, with guidance scale set to 5 and a mimic guidance scale set to 3.5. The learning rate is 1e-6 in two stages, and we optimize overall framework using Adam (Kingma & Ba, 2015).

## A.2 DESIGN OF EVALUATION METRICS

- **Portrait Animation Quality.** These metrics measure how well the generated animations capture natural motion, head movement diversity, and lip synchronization.

  *Perspective Diversity Score ($\Delta P$)* measures head motion intensity by calculating the average pose angle differences (yaw, pitch, roll) between adjacent frames. It provides an indication of the overall diversity and range of head movements generated in the video. Higher $\Delta P$ scores reflect the ability to generate a wider range of head motions, indicating flexibility in animation.

  *Sync-C* and *Sync-D* measures the synchronization between audio input and lip movements in the generated videos. Higher Sync-C scores and lower Sync-D scores indicate better alignment with the audio, reflecting natural and accurate lip movements.

- **Video Quality.** These metrics assess the technical and aesthetic quality of the generated videos, focusing on aspects like facial structure similarity and overall visual appeal.

  *Dover Score* represents the overall quality of the generated video, considering both technical aspects (sharpness, frame continuity) and aesthetic elements (visual appeal). A higher Dover score indicates superior video quality, reflecting a smooth, sharp, and visually pleasing animation.

- **Identity Preserving Ability.** These metrics evaluate how well the model retains the facial identity of the subject across various frames and conditions.

  *Face Similarity* uses a face recognition model (*e.g.*, VGG Simonyan & Zisserman (2015), FaceNet (Schroff et al., 2015), ArcFace (Deng et al., 2019)) to compute the cosine similarity between the generated faces and the reference image's face across frames. A higher score indicates consistent preservation of the subject's identity, even when generating faces from different perspectives.

  *CLIP-I Score* also serves to measure facial feature similarity across frames, ensuring that the generated faces align with the reference identity. It measures the structural similarity between the face in the reference image and the faces in each frame of the generated video using a CLIP encoder pretrained on the LAION-Face dataset. A higher CLIP-I score indicates that the facial structure in the generated video remains consistent with the reference image, contributing to higher video quality.

### A.2.1 FACE PERSPECTIVE DIVERSITY SCORE ($\Delta P$)

**Head Pose Estimation.** For each frame $i$ in a video sequence of length $N$, estimate the head pose angles (yaw, pitch, roll) using a facial landmark detection model. Let the head pose at frame $i$ be represented as a tuple:

$$\mathbf{P}_i = (\text{yaw}_i, \text{pitch}_i, \text{roll}_i)$$

---

[1] https://huggingface.co/runwayml/stable-diffusion-v1-5

[2] https://civitai.com/models/4201/realistic-vision-v60-b1

[3] https://civitai.com/models/105935/twing-shadow

**Pose Angle Differences.** For each consecutive pair of frames $(i, i+1)$, compute the absolute differences in yaw, pitch, and roll:

$$\Delta\text{yaw}_i = |\text{yaw}_{i+1} - \text{yaw}_i|$$

$$\Delta\text{pitch}_i = |\text{pitch}_{i+1} - \text{pitch}_i|$$

$$\Delta\text{roll}_i = |\text{roll}_{i+1} - \text{roll}_i|$$

**Average Pose Difference for Each Pair of Frames.** Calculate the average of these differences for each frame pair:

$$\Delta P_i = \frac{\Delta\text{yaw}_i + \Delta\text{pitch}_i + \Delta\text{roll}_i}{3}$$

**Overall Perspective Diversity Score ($\Delta$P).** The final Perspective Diversity Score is the mean of all pose differences across the video:

$$\Delta P = \frac{1}{N-1} \sum_{i=1}^{N-1} \Delta P_i$$

### A.2.2 FACE SIMILARITY METRICS

**FaceSim (Face Similarity) Calculation.** FaceSim measures the cosine similarity between the generated faces and the reference face across frames. The cosine similarity between two face embeddings $\mathbf{f}_{\text{gen}}$ (generated face) and $\mathbf{f}_{\text{ref}}$ (reference face) is given by:

$$\text{FaceSim} = \frac{\mathbf{f}_{\text{gen}} \cdot \mathbf{f}_{\text{ref}}}{\|\mathbf{f}_{\text{gen}}\|\|\mathbf{f}_{\text{ref}}\|},$$

where $\mathbf{f}_{\text{gen}}$ and $\mathbf{f}_{\text{ref}}$ are the feature vectors representing the generated and reference faces, respectively. $\|\mathbf{f}\|$ represents the norm (magnitude) of the feature vector $\mathbf{f}$. This cosine similarity score will be between $-1$ and $1$, where a higher score indicates greater similarity between the generated and reference faces.

**CLIP-I Score Calculation.** CLIP-I measures the structural similarity between the reference face and the generated faces across frames, using embeddings from the CLIP model. The cosine similarity for CLIP-I is calculated similarly:

$$\text{CLIP-I} = \frac{\mathbf{e}_{\text{gen}} \cdot \mathbf{e}_{\text{ref}}}{\|\mathbf{e}_{\text{gen}}\|\|\mathbf{e}_{\text{ref}}\|},$$

where $\mathbf{e}_{\text{gen}}$ is the embedding of the generated face from the CLIP model, and $\mathbf{e}_{\text{ref}}$ is the embedding of the reference face from the CLIP model. A higher CLIP-I score reflects better structural alignment between the reference and generated faces.

## B PORTRAIT ANIMATION PARADIGM COMPARISON

**ReferenceNet-based Framework.** The main feature of ReferenceNet-based framework is its "*Mutual Self-Attention*" mechanism within the Reference U-Net block (Chang et al., 2023; Xu et al., 2024c; Hu, 2024), as shown in Fig. 9 right. This structure tightly couples the reference image with the generated animation, within the following workflow:

- A reference image and video frames are passed through a VAE encoder, which encodes them into latent features.

- Latent of the reference image is processed by the Reference U-Net and transmitted to the Denoising U-Net to interact with the latents of the video frames.

- The key process happens in the Denoising U-Net Block where *Mutual Self-Attention* replaces standard spatial self-attention in the denoising U-Net, combining various features like audio cross-attention, and motion module features.

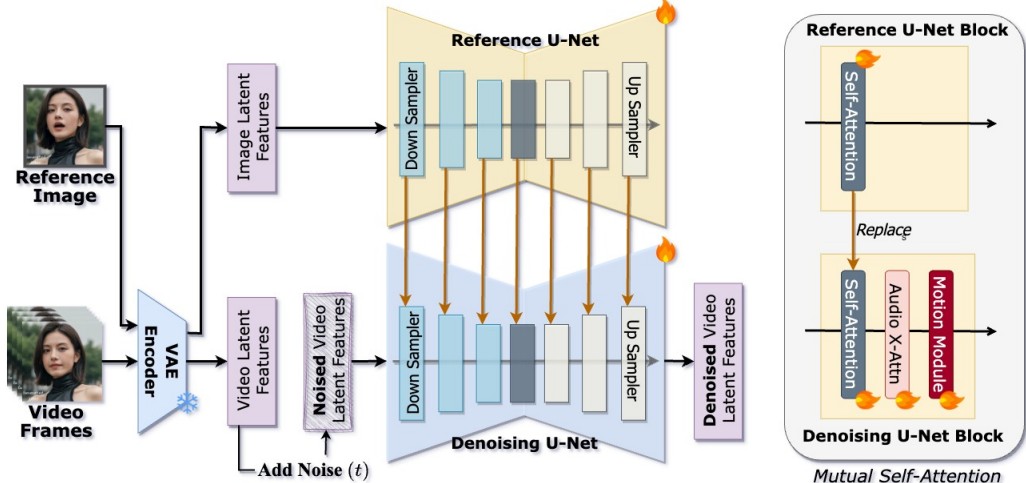

Figure 9: **ReferenceNet-based framework**. The ReferenceNet framework generates video animations by tightly coupling a reference image and video frames through a Denoising U-Net. It uses *Mutual Self-Attention* to integrate audio, motion, and identity controls, ensuring consistency but limiting flexibility in pose and background changes. Strong control over reference images drives the animation process.

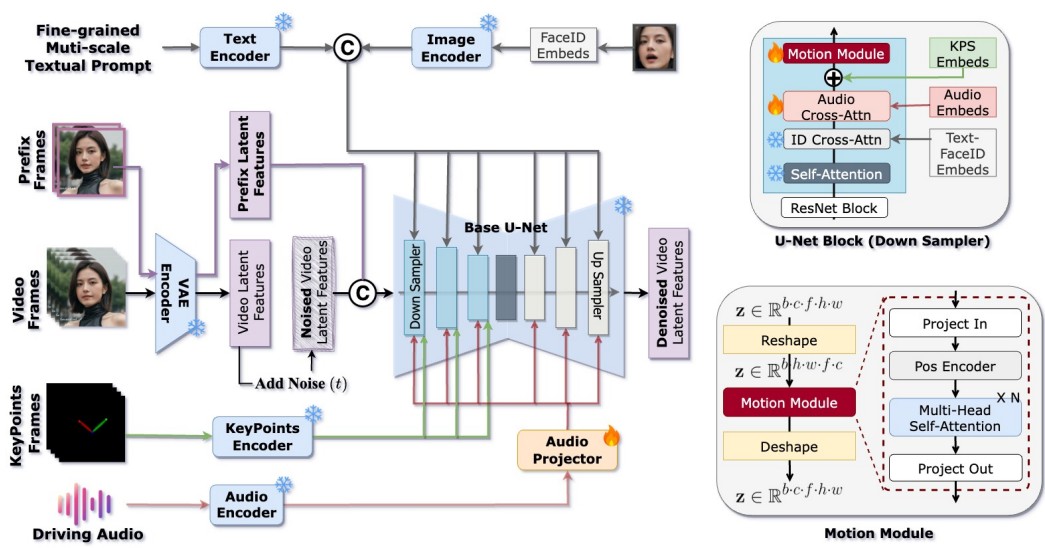

Figure 10: **AnyExpress framework**. AnyExpress eliminates the ReferenceNet, while a modular *Audio-Motion Adapter* allows flexible animation with any face pose, animated backgrounds, and text-based controls, offering more versatility and reduced training complexity.

The denoising process is heavily guided by the reference image, making the generated animation strongly constrained by the reference, which limits flexibility in pose and background changes.

**Detailed Framework of *AnyExpress*.** Fig. 10 showcases a detailed framework of *AnyExpress* compared to Fig. 2. In this version, we adapt the T2I-Adapter for face pose control, demonstrating AnyExpress's ability to freely control face positions and angles. Unlike the tightly coupled, specialized ReferenceNet, AnyExpress offers a modular design with interchangeable components (*e.g.*, Audio-Motion Adapter, Keypoints Encoder), enabling greater flexibility and adaptability.

---

**Algorithm 1** Progressive Prefix Conditioning for Long Video Generation.

---

**Input**: $\mathbf{z}_i^j$: The latent feature of $j$-th frame in $i$-th video window; $\mathbf{c}$: combinations of control conditions; $O$: the number of overlapped frames; $\mathbf{z}_{\text{anchor}}$: Anchor latents (first window) for setting statistics (mean, variance).

**Output**: $\mathbf{z}'$: A long sequence of latent features of video frames.

1: $\mathbf{z} \sim \mathcal{N}(0, I)$; {*Random initialization of video latent features.*}
2: **for** $i = 0, 1, 2, \ldots$ **do**
3:    **if** $i \neq 0$ **then**
4:       Initialize $\mathbf{z}_i^j$ with prefix frames from the last $O$ frames of window $i - 1$.
5:    **end if**
6:    **for** $t = T$ to $1$ **do**
7:       **if** $i = 0$ **then**
8:          $\mathbf{z}_i^j \leftarrow \text{DM}(\mathbf{z}_i^j, \mathbf{c}, t)$ {*Denoise the first (anchor) window.*}
9:          **Store** mean and variance of $\mathbf{z}_i^j$ as $\mathbf{z}_{\text{anchor}}$ for future windows.
10:       **else**
11:          $\mathbf{z}_i^j \leftarrow \text{DM}(\mathbf{z}_i^j, \mathbf{c}, t)$ {*Denoise each subsequent window.*}
12:          Align $\mathbf{z}_i^j$ with $\mathbf{z}_{\text{anchor}}$ using stored mean and variance. {*Anchor alignment for consistency.*}
13:       **end if**
14:    **end for**
15: **end for**
16: **return** $\mathbf{z}' = \text{Merge}(\mathbf{z})$; {*Merge latents across all windows refer to Algo. 2.*}

---

## C ALGORITHM

### C.1 PROGRESSIVE PREFIX CONDITIONING

---

**Algorithm 2** Merge algorithm for combining overlapped video windows

---

**Input:** $\mathbf{z}$: 2D list of video windows, $O$: Number of overlapping frames
**Output:** $\mathbf{z}_p$: Merged list of video windows

1: $\mathbf{z}_p \leftarrow \mathbf{z}[0]$ {*Initialize with the first video window.*}
2: **for** $i \leftarrow 1$ to $|\mathbf{z}| - 1$ **do**
3:    $\mathbf{z}_p \leftarrow \mathbf{z}_p \cup \mathbf{z}[i][O :]$ {*Extend by non-overlapping part of window.*}
4: **end for**
5: **return** $\mathbf{z}_p$

---

Algorithm 1 outlines the *Progressive Prefix Conditioning* strategy for long video generation. This method generates video frames in windows, ensuring that frames in later windows are conditioned on the "prefix frames" from the previous window. Specifically:

- For each window $\mathbf{z}_i^j$, the first $O$ frames are initialized with the last $O$ frames of the preceding window (except the first window, whose latents are all randomly initialized).

- In the first window, the latent features are denoised, and their mean and variance are stored as an *anchor* ($\mathbf{z}_{\text{anchor}}$).

- For subsequent windows, the denoising process aligns the latent features with the stored *anchor* statistics to ensure consistency.

- Finally, the latents from each window are merged into a continuous sequence.

## D EXTENDED EXPERIMENTAL RESULTS

The Extended Experimental Results section highlights the scalability and adaptability of AnyExpress across various models and core tasks. It validates that AnyExpress integrates seamlessly with different Text-to-Image models (D.1) and external adapters like ControlNet (D.2), while maintaining precise control and identity consistency. Moreover, it reinforces the framework's flexibility in handling the

three key tasks of *Freeform Portrait Animation*: generating diverse face poses (D.3), integrating dynamic animated backgrounds (D.4), and accurately following text-based control (D.5).

## D.1 COMPATIBILITY WITH PERSONALIZATION BASE MODELS

We evaluate how well AnyExpress integrates with various Text-to-Image (T2I) personalization models, ensuring that the flexibility of the AnyExpress framework does not compromise the quality or personalization of generated content. The evaluation is split into two key aspects: generating animations with a reference image and generating animations solely based on textual descriptions.

Fig. 11 showcases the results of various T2I models when using a reference image to generate different animation styles such as Photorealism (`Realistic V6`), Cartoonish (`ChunkyCat`[4]), Semi-Realism (`LusterMix v15`[5]), and Digital Paint (`Toonyou beta6`[6]). It demonstrates that AnyExpress is capable of maintaining the identity of the subject across different animation styles while adapting to the stylistic variations introduced by each personalized model.

Fig. 12 expands on this by removing the reference image and instead generating animations based on detailed text descriptions, with extended personalized models[7][8][9][10]. This demonstrates the framework's flexibility in handling both identity and background details (*e.g.*, "flames swirling inside a fireplace" vs. "cars moving along a highway"). The results indicate that the personalization base models can interpret textual descriptions to generate visually distinct outputs while maintaining consistency with the text-based identity information.

## D.2 COMPATIBILITY WITH CONTROLNET

We examine how the AnyExpress framework integrates with ControlNet to enhance controlled generation through the use of face landmark sequences. The primary goal is to determine how well AnyExpress can maintain identity consistency, lip synchronization, and facial expressions while employing other off-the-shelf adapters for controlling the driven videos.

Fig. 13 and 14 demonstrate the compatibility of AnyExpress with ControlNet, using landmark sequences to guide the facial expressions of various animated subjects. The driven videos (*e.g.*, Mona Lisa and Joker) show how facial landmarks are transferred across different subjects, retaining their identity and lip-sync precision. These results validate that ControlNet can be effectively integrated with AnyExpress to produce controlled animations with high consistency across various facial poses, identities, and styles, demonstrating the flexibility of AnyExpress in controlled settings.

## D.3 EXTENDED RESULTS ON ANY FACE POSE

We further explore the ability of AnyExpress to generate diverse facial orientations and movements while preserving identity consistency across different scenarios. The evaluation is performed through various comparisons with baseline methods, different identities, and control signal scaling.

**Further Comparison on Any Face Pose.** Fig. 15 compares AnyExpress with baseline methods following the same setting as Fig. 5 . The results show that AnyExpress outperforms the baselines in preserving facial identity and following the guidance signals accurately across varying face poses.

**Robust Identity Preservation.** Fig. 16 further demonstrates AnyExpress's ability to maintain identity consistency while following the same face guidance signals across multiple subjects. This robustness across various identities proves that AnyExpress can handle a wide range of facial characteristics and styles without losing the unique aspects of each individual face.

**Face Control Signal Intensity.** Fig. 17 focuses on scaling the KeyPoints control signals to adjust facial expressions and orientations from scale 0.6 to 1.1. The results show that AnyExpress is scalable

---

[4]`https://huggingface.co/Yntec/ChunkyCat`
[5]`https://civitai.com/models/85201/lustermix`
[6]`https://civitai.com/models/30240/toonyou`
[7]`https://civitai.com/models/43331/majicmix-realistic`
[8]`https://huggingface.co/Yntec/AnimephilesAnonymous`
[9]`https://huggingface.co/Yntec/Genesis`
[10]`https://huggingface.co/Yntec/GrandPrix`

and adapts well to different control signal intensities, maintaining facial coherence and dynamic movements without introducing distortions or identity misalignments.

### D.4 Extended Results on Any Animated Contexts

Extending from Fig. 6a, we further demonstrate the ability of AnyExpress to integrate dynamic animated backgrounds into generated videos in Fig. 18, showcasing its flexibility beyond static settings. The reference images are paired with various animated background scenes, including sails billowing, waves crashing, volcano eruption, flames dancing, and rivers flowing. AnyExpress adapts seamlessly to each background while maintaining the identity and expression of the subjects, indicating its robustness in handling complex animated environments without sacrificing facial identity or motion synchronization.

### D.5 Extended Results on Any Text Control

Extending from Fig. 6b, we further demonstrate how AnyExpress generates animations based on textual descriptions for both the subject's identity and the animated background, without relying on reference images. In Fig. 19 and 20, these texts include details on the subject's physical features (*e.g.*, hair color, skin tone, clothing) and the background scene (*e.g.*, flames swirling, waves crashing, birds taking flight). These results confirm AnyExpress's ability to effectively translate complex textual prompts into visually distinct animations, ensuring consistency in both the identity and the corresponding background as described in the text.

## E  Limitations and Future Work

Since *AnyExpress* relies on a highly scalable and modular *audio-motion adapter*, it provides a flexible foundation that can easily integrate with more advanced models and techniques in future work. This modularity ensures that only lightweight modules are trained, allowing for seamless adaptation to new developments without the need for extensive re-training. This scalability makes *AnyExpress* ideal for incorporating state-of-the-art models and methodologies. As the field of diffusion models continues to evolve, this architecture can be enhanced with new capabilities while maintaining efficiency and adaptability. These following aspects underscore where future research can refine and augment the methodology presented in *AnyExpress*: **(1) Incorporating Advanced Base Models:** The architecture of *AnyExpress* is currently built on SD1.5. However, emerging advanced models like SDXL, DiT, SD3, and FLUX offer enhanced capabilities for more complex tasks, holding the potential to significantly improve the quality and flexibility of portrait animation. **(2) Enhancing Identity Control:** Integrating advanced identity controllers like PuLID (Guo et al., 2024c) and MoMA (Song et al., 2024) can improve identity consistency. Techniques like InstantID (Wang et al., 2024b), which offers finer identity control, are promising, though currently incompatible due to the difference in base models (SDXL vs. SD1.5). Resolving these compatibility issues would unlock significant gains in identity preservation. **(3) Improved Audio-Visual Synchronization:** Incorporating more sophisticated synchronization techniques, such as advanced audio analysis and cross-modal learning, could further enhance the alignment of facial movements with audio, especially in nuanced or emotionally expressive contexts. **(4) Enhancing Temporal Coherence:** Advanced temporal coherence mechanisms are required to address inconsistencies in fast or intricate sequences. Leveraging long-term dependencies or recurrent neural networks could help achieve smoother transitions and eliminate flickering. **(5) Boosting Computational Efficiency:** Optimizing computational efficiency and reducing the steps (Song et al., 2023; Luo et al., 2023; Wang et al., 2023) would make *AnyExpress* more suitable for real-time applications.

## F  Broader Impacts

There are broader social implications tied to the development of portrait animation technologies, particularly when driven by audio inputs. One significant concern is the potential misuse of such technologies for deceptive purposes, including deepfakes. This poses ethical risks related to the creation of highly realistic, animated portraits that could be used for malicious intent. To mitigate these risks, it is crucial to establish clear ethical guidelines and promote responsible usage of the technology.

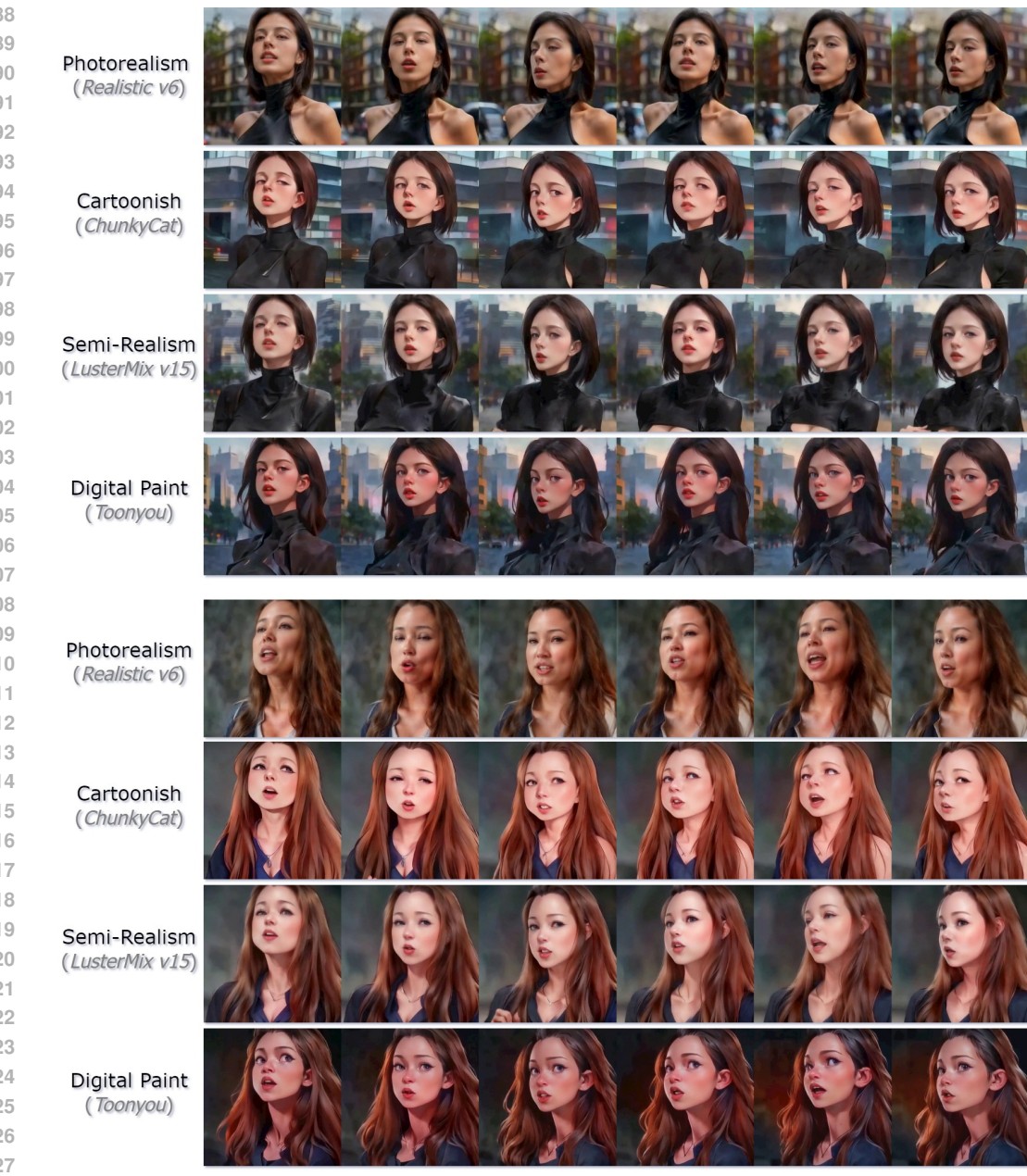

Figure 11: Results from **various personalized T2I models** using a reference image. The models—Realistic v6, ChunkyCat, LusterMix v15, and Toonyou—demonstrate Photorealism, Cartoonish, Semi-Realism, and Digital Paint styles, respectively, while maintaining the subject's identity across frames.

Furthermore, issues around privacy and consent must be addressed, especially regarding the use of individuals' likenesses and voices in animated outputs. Ensuring transparent data policies, obtaining informed consent, and protecting individuals' privacy rights are essential steps. By proactively tackling these challenges, *AnyExpress* aims to contribute to the ethical and responsible advancement of portrait animation technology within society.

Figure 12: Results from **various personalized T2I models generated from textual descriptions**. The identity and background are described purely through text, with different models interpreting the descriptions to generate visually distinct yet consistent outputs.

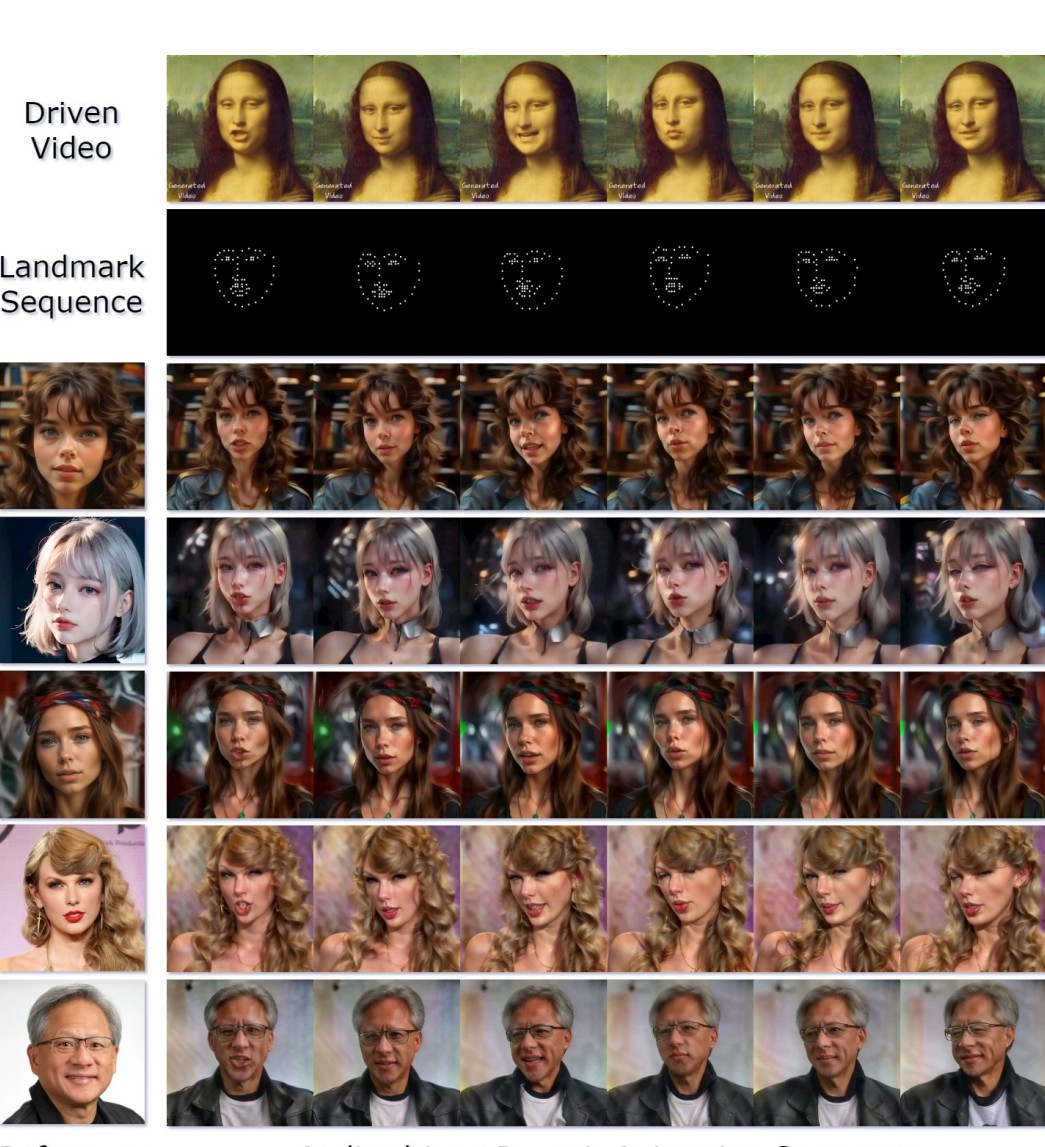

**Driven Video**

**Landmark Sequence**

**Reference**  Audio-driven Portrait Animation Sequence

Figure 13: **Face landmark sequence with ControlNet**, showing audio-driven portrait animation using *Mona Lisa* as the driven video.

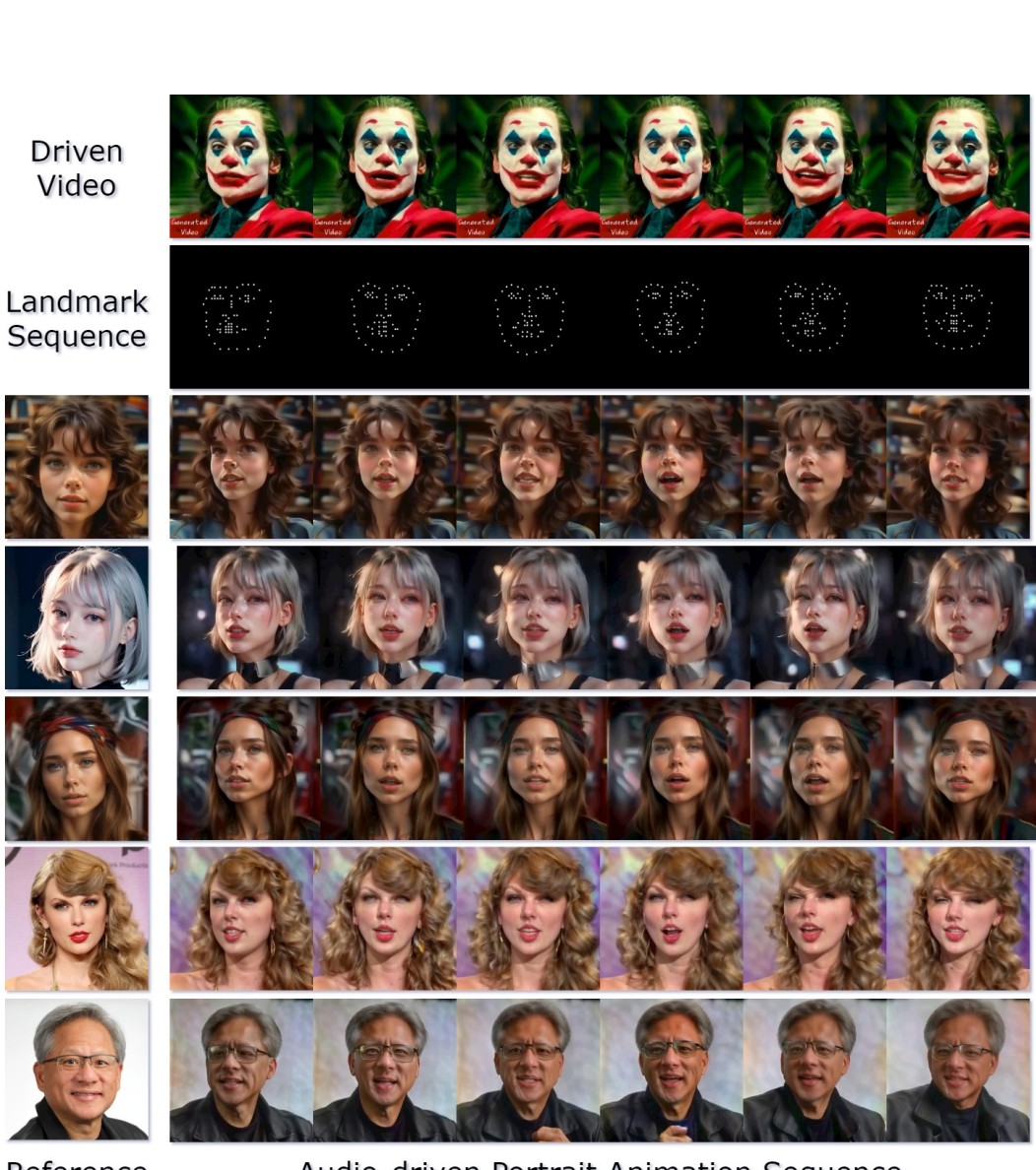

Figure 14: **Face landmark sequence with ControlNet**, showing audio-driven portrait animation using the *Joker* as the driven video.

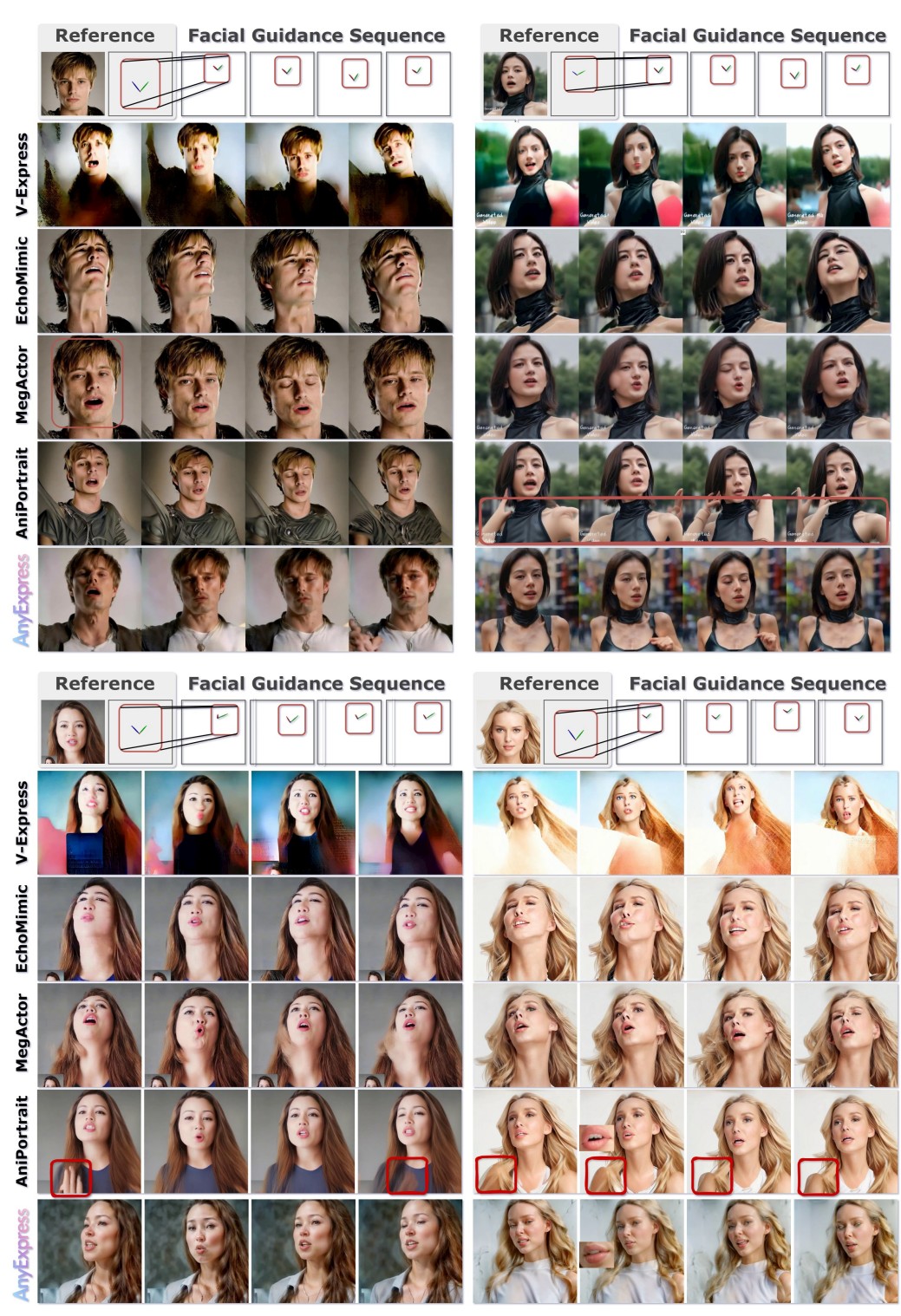

Figure 15: Extended comparison between AnyExpress and baseline methods on **Any Face Pose** generation.

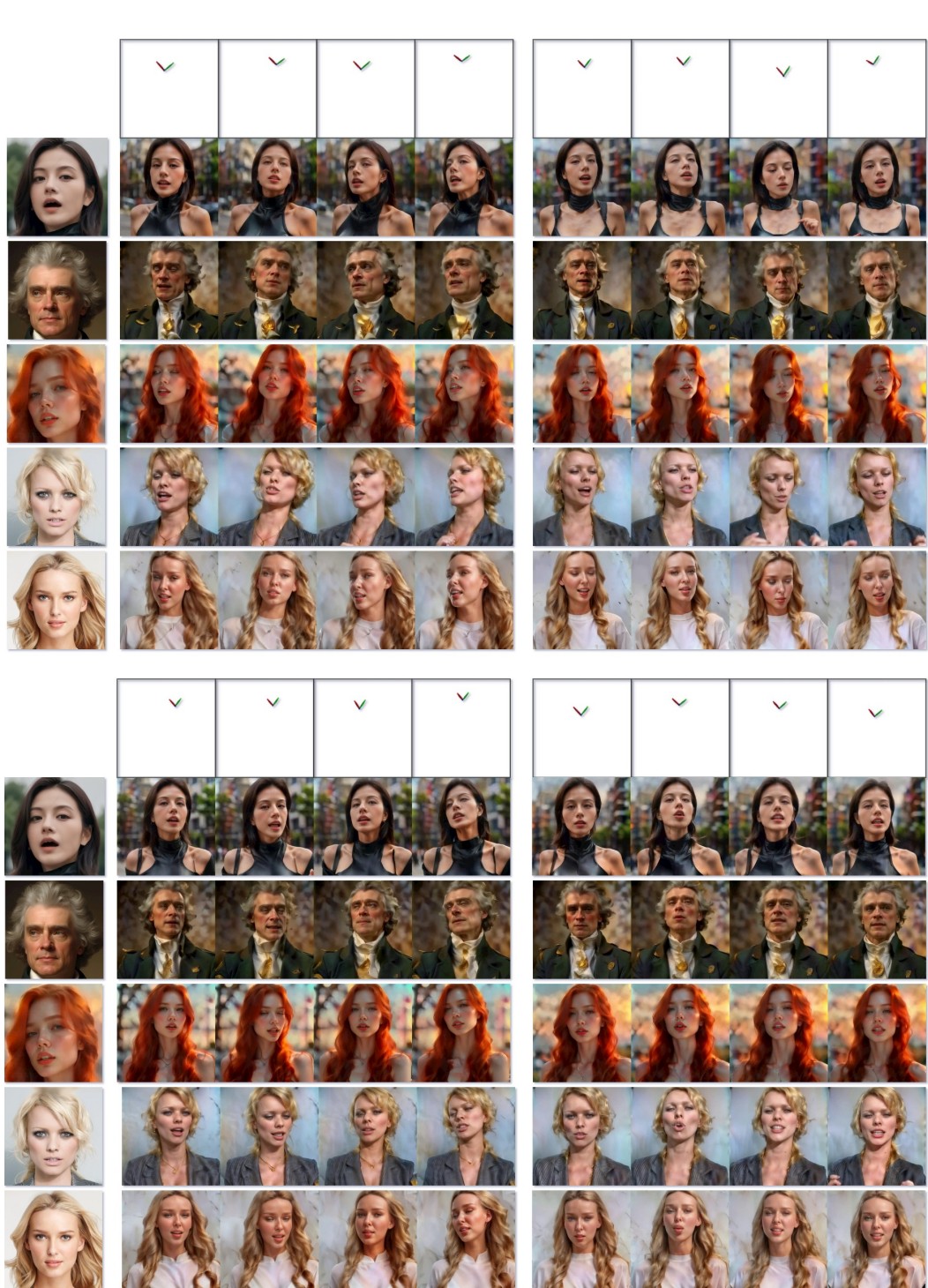

Figure 16: Adaptability of AnyExpress to various identities with the **same facial guidance signals**. Across a range of different facial characteristics, the identity consistency is maintained while accurately following the given pose and expression transitions.

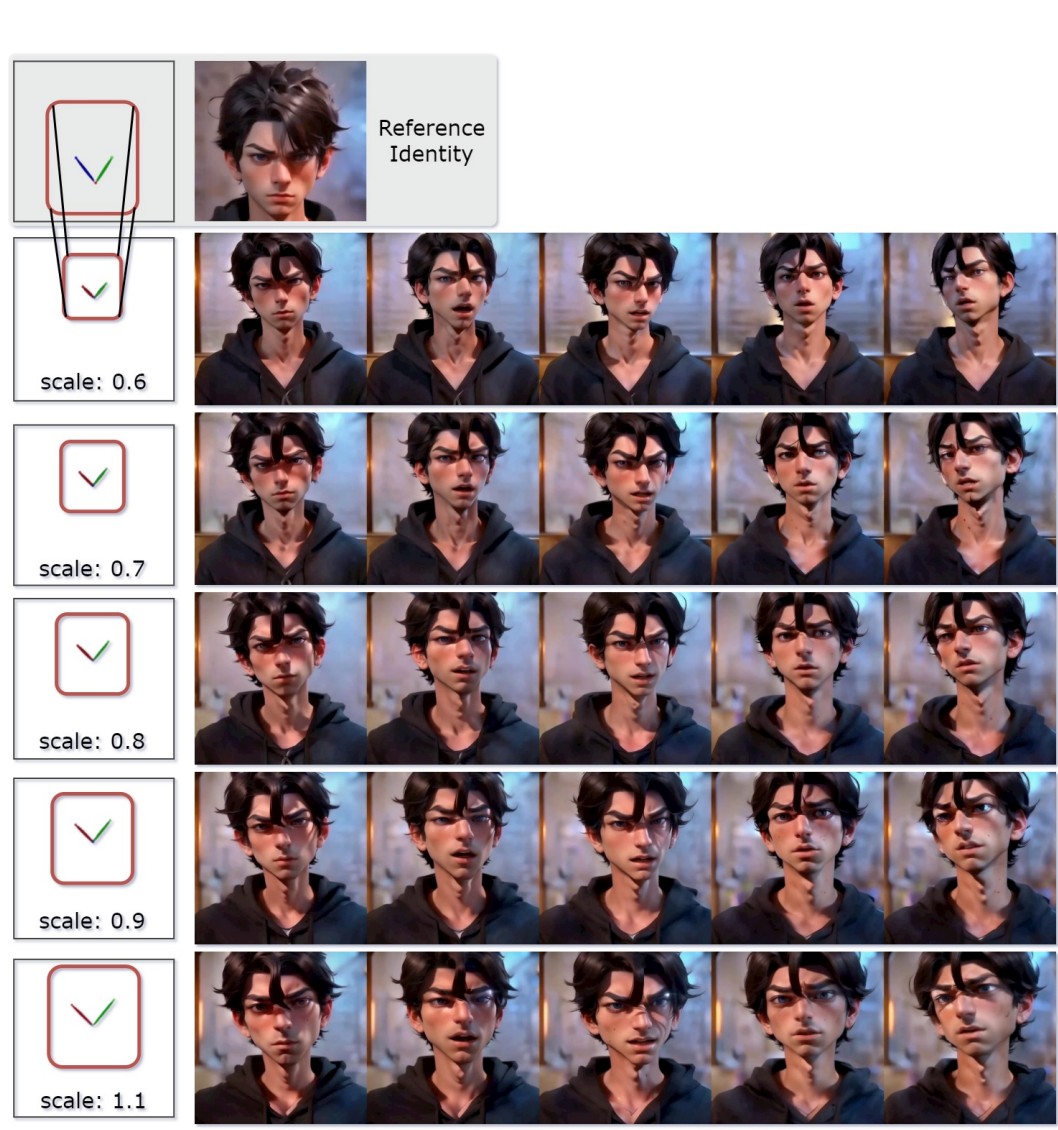

Figure 17: Scalability of AnyExpress by adjusting the KeyPoints control signals from scale 0.6 to 1.1.

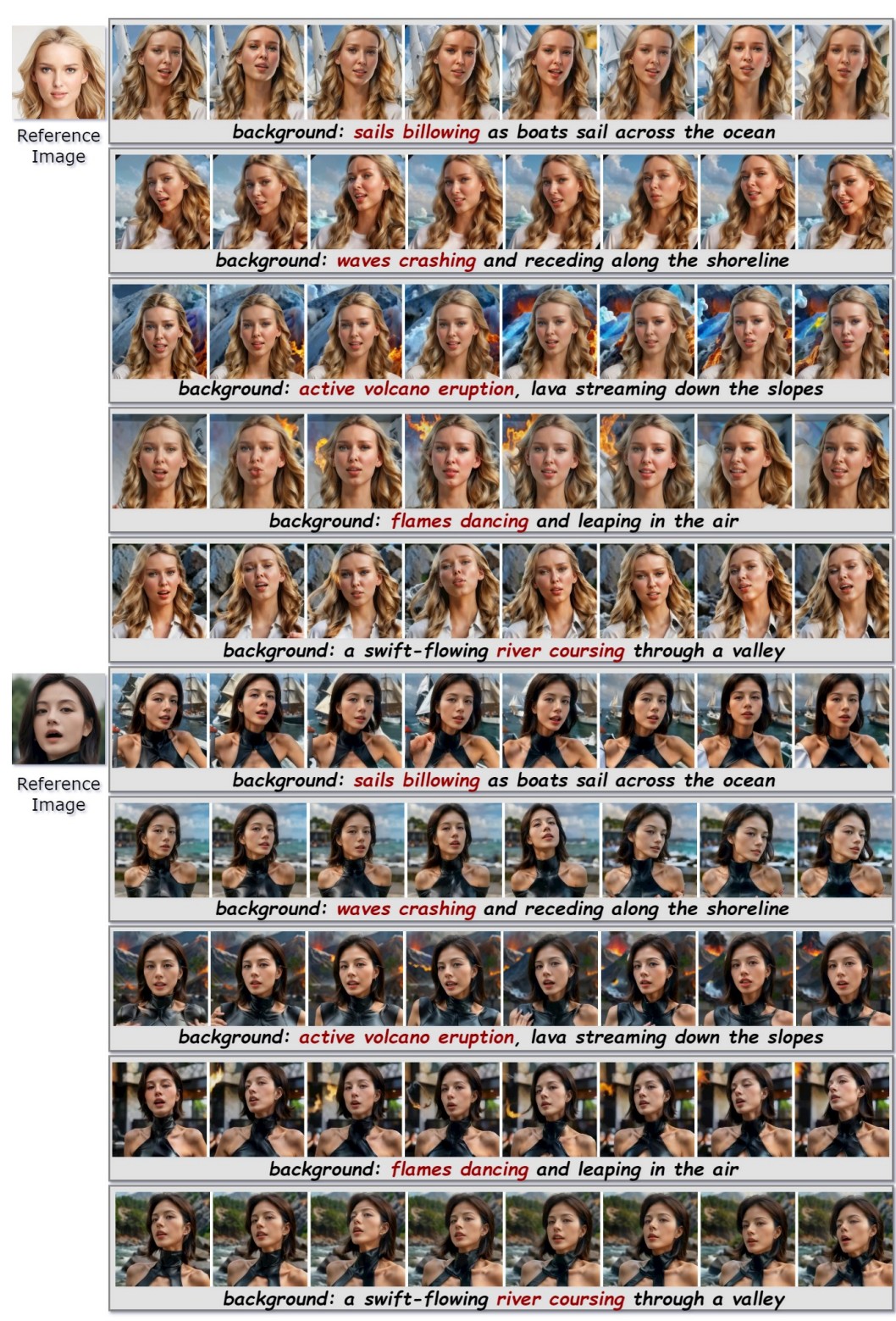

Figure 18: AnyExpress integrates **dynamic animated backgrounds** with consistent subject identity and motion. The backgrounds vary from sails billowing and waves crashing to volcano eruptions and flames dancing, showcasing the flexibility of AnyExpress in diverse animated contexts.

Figure 19: AnyExpress generates **animations based on textual descriptions** of identity and background.

Figure 20: AnyExpress generates **animations based on textual descriptions** of identity and background.

