# OpenReview forum: "AnyExpress: One Adapter Enabling Highly Flexible Audio-Driven Portrait Animation"
_ICLR.cc/2025/Conference — ICLR 2025 Conference Withdrawn Submission_

### Official Review · Reviewer_3PXi · 2024-10-21

**Soundness:** 3
**Presentation:** 3
**Contribution:** 2
**Rating:** 3
**Confidence:** 3

**Summary:**

This paper presents AnyExpress, a modular framework for audio-driven portrait animation that eliminates the reliance on ReferenceNet, reducing training complexity and enabling more flexible and expressive animations. By introducing a lightweight Audio-Motion Adapter and optimizing motion dynamics, AnyExpress achieves high flexibility and efficiency during training. The authors claim that the proposed method showed freedom in generating videos with background, lower demand, and can be used with custom models.

**Strengths:**

This paper proposes a new framework, Anyexpress, which eliminates the dependence on the large and complex ReferenceNet architecture, reducing the model's parameter count. Additionally, by training the audio-motion adapter, Anyexpress operates independently of the U-Net architecture, making it easy to integrate with various diffusion models and control adapters. The adapter framework, without requiring training of the overall model, can expand the model's utility.

**Weaknesses:**

This paper claims that weak control enables expressive and varied outputs, but the lack of precise control may limit consistency in the results. Moreover, the absence of quantitative evaluation in the ablation study limits the ability to rigorously assess the model’s performance and its proposed components. I recommend providing quantitative results for the ablation study. Additionally, no user study (perceptual study) is presented, which is crucial for evaluating naturalness of motion, identity preservation, or lip sync quality, etc as perceptual quality is a key aspect of audio-driven talking head animation research.

Furthermore, visual artifacts, such as saturated images, are still present despite the use of dynamic thresholding (Line 872). While I agree with the merit of the adapter-based audio-driven talking head animation method, the current results are lacking in consistency and visual quality.

**Questions:**

1. The paper claims the efficiency of the ReferenceNet-free architecture, but the required training time isn't provided. How long did it take to train using 8 V100 GPUs?
2. In comparison with the baselines in Figure 5, some results are worse than I expected or have experienced myself. Could this be due to the facial guidance not aligning with the reference? If yes, why are the reference images cropped at the center, while the facial guidance sequence is not?
3. In your personal opinion, do you think there is a trade-off between diversity and synchronization fidelity? In case of weak guidance?

**Details Of Ethics Concerns:**

Human subject

---

### Official Review · Reviewer_GSER · 2024-10-23

**Soundness:** 3
**Presentation:** 4
**Contribution:** 2
**Rating:** 6
**Confidence:** 4

**Summary:**

This paper examines the problem of animating a talking face using audio, an identity signal and additional optional signals. It first discusses and positions itself relative to existing works (e.g. EMO, V-Express) that use a strongly constrained heavy reference UNET architecture with a very large parameter count. This work proposes a lighter-weight adaptor that can be used with any existing text-to-image model and is less tightly coupled to input signals. This allows for easier training and a more robust model for large variations in input signals (primarily, the paper looks at head pose).

**Strengths:**

The paper's premise is that models based on a reference UNET have some limitations. I believe the authors are correct that these models are particularly limited by the tightness of their control and, therefore, cannot model significant variations in head pose. This is well-motivated, and the authors demonstrate it. In particular, Figures 3 and 5 together make this a strong case.

The quantitative and qualitative results show that the proposed model works better in this scenario. The addition of dynamic backgrounds and text control shows the model's flexibility nicely.

The paper is well written. The figures, in particular, are a delight and illustrate the main argument well.

The novelty of this work is sufficient. While many of the components are similar to existing state-of-the-art, and hijacking text-to-image models is not itself novel, the novelty is there as the model trains an adapter instead of a large reference net.

The decision to release code is great and should help drive further research in this area.

Overall, I like the flexibility of this approach even if the results are underwhelming.

**Weaknesses:**

My main concerns with this work are as follows:

First, I am unsure how applicable the use case is. In particular, the AnyPose scenario may be overly contrived. The proposed use case requires the driving head pose to differ significantly from the reference. However, I do not see why someone wishing to do this would not simply align the driving head pose to the reference image's head pose using a rigid alignment technique. Explaining the reason for needing diverse poses that cannot be met with alignment would help aid understanding of the model.

The second concern is that the overall subjective quality of the results is relatively poor, especially compared with ReferenceNet-based models, which appear an order of magnitude better than the results of AnyExpress. In particular, the model suffers from jerky movements and temporal artefacts such as hair appearing and then disappearing.

The advantage of a smaller network doesn't seem to be emphasised in this work. You still train on 8xV100 GPUs, which is not a major improvement. Also, inference times are omitted, which may be one of the most convincing arguments. It would strengthen the paper to include a more detailed analysis of the training times, hardware requirements and inference speed compared to ReferenceNet models.

There are also some other minor weaknesses.

I'm not entirely clear on which modules are frozen vs trained in the AnyNet UNET; please see Questions. An updated Figure 2 to look more like those in reference net papers, e.g. EMO (with the submodules inside the Up/Down blocks), would help. An additional diagram in the supplementary with a more detailed UNET architecture, including labels for which components are frozen/trained may also be helpful.

There are several grammatical and typographical errors, e.g.:
L110: Recent diffusion-based portrait animation methods focus -> focuses
L481: How Progressive Prefix Conditioning Maintaining Consistency? -> How Does Progressive Prefix Conditioning Maintain Consistency

These have no impact on my decision, but it would be worth a final proofreading of the paper.

**Questions:**

Can this model be trained on commercial hardware, e.g., a 30- or 40 series GPU?

How fast is the inference? For example, how long would it take to generate a 10-second video? How does this compare to the other SOTA? It would be good to include this in Table 2.

In Figure 2, you show that the UpBlock is frozen and the Adaptor is being trained. Is this adaptor applied between the up and down blocks, e.g. at the bottleneck, or is it a part of the down block? If it is at the bottleneck, is the down block also frozen? This should be reflected in the figure.

---

### Official Review · Reviewer_Go8u · 2024-11-03

**Soundness:** 3
**Presentation:** 3
**Contribution:** 2
**Rating:** 5
**Confidence:** 4

**Summary:**

The paper introduces AnyExpress, a method for portrait animation (Freeform Portrait Animation method). Unlike previous methods that depend on ReferenceNet (2D U-Net architecture), AnyExpress removes the need for these modules. Instead it uses an audio-motion adapter that reduces trainable parameters and integrates with an existing T2I SD models. AnyExpress is a multimodal method, combining novel conditioning inputs such as FaceID, text, and animation context alongside traditional controls like audio and facial pose. This enables the method to generate identity-consistent videos with dynamic backgrounds, conditioned on audio, text, and the pose.

**Strengths:**

AnyExpress has several strengths:

1) It introduces a lightweight module that allows conditioning with fewer trainable parameters compared to previous methods. The architecture first uses standard audio and text encoders. The embeddings from these encoders are sent to an audio-motion adapter, which includes an audio projector and motion modules as part of the U-Net block (downsampler), as shown in fig 10.

2) Progressive prefix conditioning with anchor alignment (shown in the last row of fig 1a) demonstrates the effectiveness of AnyExpress. The generated frames show no color fading, consistent clothing/textures, and consistentidentity.

3) AnyExpress also adds control over dynamic backgrounds, which is a novel feature.

**Weaknesses:**

1) Since the modules mentioned in Stregnths (1) are frozen, it’s critical to compare results with a baseline method where audio or text embeddings are added directly to SD using methods like concatenation or addition to the U-Net. This comparison would show the impact of using the audio motion adapter vs a simpler approach, helping to confirm whether the contribution is substantial. I request the authors to include a comparison of this baseline ablation for the metrics mentioned in tab 2.
2) The pose diversity score, Sync C, and Sync D scores are lower than other methods like AniPortrait and Echnomimic, as shown in tab 2. Similarly, AniPortrait, MegaActor, and Echnomimic generate valid portrait poses, and AniPortrait and MegaActor have quality samples similar to AnyExpress, making it hard to see the unique benefits of AnyExpress as these other methods perform better in tab 2 scores. The authors could provide a more detailed analysis explaining why AnyExpress underperforms on certain metrics despite its claimed advantages.
3) There are no ablation quantitative scores to validate the impact of different components in the proposed architecture. The authors should provide a quantitative analysis with ablation components, including those mentioned in Weakness (1), Progressive Prefix Conditioning, varying the number of trainable blocks in the motion module, and the used frozen models for audio/ text/ image/ keypoint encodings.
4) Also, because of the various conditioning factors, it’s challenging to assess the quality of individual elements in the generated outputs.
5) Without video, it’s impossible to judge lip sync from just frames. The authors should provide video samples in supplementary materials and potentially include quantitative metrics specifically designed to evaluate lip sync quality over time.
6) The generated backgrounds and identity frames show low quality in fig 5 and 6, with inconsistent quality across complex backgrounds. It would be helpful for the authors to provide a more detailed comparison of background quality across methods, including examples with plain backgrounds for comparison with previous methods. Additionally, objective metrics for assessing background quality and consistency would strengthen the evaluation.
7) Subjects in the videos appear smudged. Recent work like SVP (https://arxiv.org/abs/2409.18083), which uses T2I with 3D Morphable Face Models, could be a useful comparison. The authors could include a qualitative and quantitative comparison with SVP, focusing specifically on facial detail preservation and overall image sharpness.

**Questions:**

1) The explanation for the audio projector is unclear, and the motion module's contribution seems trivial. Details about the specific text, image, audio, and keypoint encoders used are missing.

2) In fig 7b (top row), does training only the audio model result in a consistently closed mouth?

3) L 538: The claim that Progressive Prefix Conditioning ensures smooth transitions in long video sequences—how long of a sequence can actually be generated?

4) L 80-81: The statement about the “limited generalizability of the ReferenceNet-based paradigm” is unclear. Since ReferenceNet isn't designed for text or audio conditions, generalizability could be tested by adding off-the-shelf audio or text encoders to a base ReferenceNet setup. This relates to my Weakness (1)

5) The paper mentions a flexible FaceID control signal, but it's unclear if this is a categorical embedding limited to a fixed set of identities. Fig 10 shows an input image processed to generate FaceID embeddings, which are then fed into the image encoder. Is FaceID a categorical label, and does it support only a fixed number of identities? For instance, in fig 10 (left block, top part), does this identity appear during training? More details are needed to understand its generalizability, especially for unseen/ in the wild identities.

6) What is the FPS of the generated animations?

7) What’s the reasoning behind excluding samples with significant camera movement from the dataset, considering that the method is intended to handle any angle or position?

---

### Official Review · Reviewer_Xekv · 2024-11-04

**Soundness:** 2
**Presentation:** 3
**Contribution:** 2
**Rating:** 3
**Confidence:** 5

**Summary:**

The authors developed a novel framework for free-form audio-driven portrait animation. Built upon Stable Diffusion and AnimateDiff, this framework incorporates a plug-and-play audio-motion module, enabling personalized control from T2I models. Furthermore, they employ a progressive prefix conditioning strategy to ensure temporal smoothness in the generation of long videos.

**Strengths:**

1. **The overall design is lightweight**. Compared to the Reference Net module, this work enables audio-driven portrait animation with significantly fewer trainable parameters, while identity conditioning is achieved using a frozen IP-Adapter.

2. **The plug-and-play adapter supports free-form control**. The audio-motion adapter functions similarly to AnimateDiff, providing greater versatility for combining with custom T2I models.

**Weaknesses:**

1. **Visual quality and temporal consistency**. In most of the video results, the facial motion appears unnatural, and significant artifacts remain.

2. **Audio-lip synchronization**. According to the evaluation results in Table 2, the audio-lip synchronization score is inferior to that of EchoMimic. However, this aspect is critical for audio-driven talking face synthesis.

3. **Face similarity metrics**. The face similarity scores are abnormally low (<0.5), the face ID preserving is bad.

4. **Scale of training data**. In contrast to the lightweight network, the training data still comprises 300 hours of video, which is comparable to EMO's 250 hours. The very limited computing resources (8 V100 GPUs) do not support such a large scale of data effectively.

While a lightweight and versatile framework is advantageous, sacrificing quality too much undermines its value. Ensuring high-quality output should remain a priority, as a system that compromises on visual fidelity is not worth the trade-offs.

**Questions:**

See weaknesses (3 and 4 highlighted)

---

### Note · Authors · 2024-11-21

**Comment:**

We sincerely appreciate the reviews provided. While we have decided not to not move forward with this version of our submission, we are committed to addressing the points raised and continuing to improve.

Thank you once again for your time and effort.

**Withdrawal Confirmation:**

I have read and agree with the venue's withdrawal policy on behalf of myself and my co-authors.